# Fine-mapping a genome-wide meta-analysis of 98,374 migraine cases identifies 181 sets of candidate causal variants

Heidi Hautakangas [1] ✉, Joonas Kartau [1], FinnGen*, Aarno Palotie [1,2,3], Matti Pirinen [1,4,5] ✉ & International Headache Genetics Consortium*

Migraine is a highly prevalent neurovascular disorder for which genome-wide association studies (GWAS) have identified over one hundred risk loci, yet the causal variants and genes remain mostly unknown. Here, we meta-analyze three migraine GWAS including 98,374 cases and 869,160 controls and identify 122 independent risk loci of which 35 were new. Fine-mapping of a meta-analysis is challenging because some variants may be missing from some participating studies and accurate linkage disequilibrium (LD) information of the variants is often not available. Here, using the exact in-sample LD, we first investigate which statistics could reliably capture the quality of fine-mapping when only reference LD is available. We observe that the posterior expected number of causal variants best distinguishes between the high- and low-quality results. Next, we perform fine-mapping for 102 autosomal risk regions using FINEMAP. We produce high-quality fine-mapping for 93 regions and define 181 distinct credible sets. Among the high-quality credible sets are 7 variants with very high posterior inclusion probability (PIP > 0.9) and 2 missense variants with PIP > 0.5 (rs6330 in *NGF* and rs1133400 in *INPP5A*). For 35 association signals, we manage to narrow down the set of potential risk variants to at most 5 variants.

Migraine is a common neurological disorder characterized by recurrent disabling episodes of severe headache that are typically one-sided, pulsating in nature, and accompanied by other symptoms such as nausea, and hypersensitivity to light and/or sound. It has two main subtypes, migraine without aura and migraine with aura. The aura is a reversible visual, sensory, or speech disturbance that typically occurs before the headache phase. Migraine attacks last usually from 4 to 72 h, and can significantly harm daily life of patients[1]. Migraine was ranked as the second most disabling disease worldwide in terms of years lived with disability by the Global Burden of Diseases Study in 2019[2]. Its lifetime prevalence has been estimated to be about 15 to 20 %

worldwide, and it is three times more common in females than in males[2]. Family and twin studies estimate the heritability to be about 40%[3]. To date, over 100 migraine-associated loci have been reported by GWAS[4–14]. The genetic association of migraine has shown a general enrichment in genes highly expressed in vascular and central nervous system-related tissues[13,15] but we lack detailed information on specific genetic variants that affect the migraine risk.

Identification of causal genes and variants that have a biological effect on migraine is crucial for understanding the biology of migraine and for developing new effective treatments for the disorder. Here, we aim to narrow down correlated genetic variation in migraine-

[1]Institute for Molecular Medicine Finland (FIMM), Helsinki Institute of Life Science (HiLIFE), University of Helsinki, Helsinki, Finland. [2]Analytic and Translational Genetics Unit, Department of Medicine, Department of Neurology and Department of Psychiatry Massachusetts General Hospital, Boston, MA, USA. [3]The Stanley Center for Psychiatric Research and Program in Medical and Population Genetics, The Broad Institute of MIT and Harvard, Cambridge, MA, USA. [4]Department of Public Health, University of Helsinki, Helsinki, Finland. [5]Department of Mathematics and Statistics, University of Helsinki, Helsinki, Finland. *Lists of authors and their affiliations appear at the end of the paper. ✉e-mail: heidi.hautakangas@helsinki.fi; matti.pirinen@helsinki.fi

associated regions to a smaller number of candidate causal variants by applying statistical fine-mapping[16]. Fine-mapping methods evaluate how plausibly each variant in the region is among the causal variants by utilizing the observed association statistics and the LD structure of the region[16]. Multiple methods that can utilize GWAS summary statistics have been developed, including PAINTOR[17], CAVIAR[18], FINEMAP[19], JAM[20], and SuSIE[21]. The optimal way to apply fine-mapping is to compute the LD information from the original GWAS data (in-sample LD), but when the original genotype data are unavailable, approximate LD information is often obtained from a reference genotype panel (reference LD). However, when reference LD is used, the discrepancy from the in-sample LD can cause errors in fine-mapping, and this problem becomes more severe as the GWAS sample size grows[22].

Even though large meta-analyses have become a successful way to increase the statistical power of GWAS, they remain difficult to fine-map reliably for several reasons[23]. First, meta-analyses are combinations of multiple studies, and typically, no single analyst has access to the exact in-sample LD of the whole meta-analysis, which means that reference LD must be used. Second, differences in genotyping platforms and genotype imputation pipelines between the meta-analyzed studies can bias the fine-mapping results. Third, some variants included in the meta-analysis may be present in only a subset of the studies, which leads to variation in the information content of the association statistics of different variants. In a landmark fine-mapping study on schizophrenia, Trubetskoy et al. (2022)[24] avoided these problems by collecting all genotype-phenotype data into a single analysis site. Unfortunately, to our knowledge, no other international disease consortium has been able to create a comparable analysis environment that would allow an in-sample fine-mapping of a large meta-analysis. Given that fine-mapping of meta-analysis results typically relies on reference LD, a crucial question is how we can assess when the results of fine-mapping based on reference LD are reliable.

So far, the largest GWAS meta-analysis on migraine contained 102,084 cases and 771,257 controls from 25 study collections[13]. Unfortunately, we cannot perform reliable fine-mapping for that meta-analysis, since the in-sample LD is not available. Instead, we conducted a migraine meta-analysis with 98,374 migraine cases and 869,160 controls of European genetic ancestry by combining data from three sources: 23andMe, Inc., FinnGen and UK Biobank (UKB). Of these data sets, 23andMe and UKB were included in the earlier meta-analysis of Hautakangas et al. (2022) while FinnGen was not. Statistical power of our meta-analysis was comparable to the previous migraine meta-analysis of Hautakangas et al. (2022), with effective sample sizes of 339,000 and 326,000, respectively. Importantly, we have the full in-sample LD available for 26 previously known migraine risk loci whereas for the remaining risk loci we have access to the in-sample LD for FinnGen and UKB but not for 23andMe (Table 1). This set-up allowed us to investigate how different LD reference panels perform compared to the in-sample LD. In particular, we evaluated different statistics that could be used to assess fine-mapping quality when only reference LD is available. Finally, we utilized our results to fine-map 102 migraine risk loci to narrow down the putative causal variants behind the associations.

In this work, we identify 122 susceptibility loci for migraine of which 35 have not been reported before[4–14]. We observe that in fine-mapping with reference LD, the posterior expected number of causal variants (PENC) is most suitable to classify the high- and low-quality results. We are able to get reliable fine-mapping results for 93 out of 102 regions for migraine, and identify 7 variants with a high probability (>90%) of being causal, and two missense variants, rs6330 in *NGF* and rs1133400 in *INPP5A*, with a probability > 50% of being causal. Our study demonstrates that, with a suitable analysis approach, the data sets already available are sufficient to yield reliable and new fine-mapping results.

## Results

We conducted an inverse-variance weighted meta-analysis on migraine by combining results from the three GWAS (Table 1): UK Biobank (UKB; 10,881 cases and 330,169 controls), 23andMe, Inc. (53,109 cases and 230,876 controls), and FinnGen Release 8 (34,385 cases and 308,114 controls). The total sample size is 98,374 migraine cases and 869,160 controls. Before meta-analyzing the data, we estimated pairwise genetic correlations between the study collections by LD Score regression (LDSC)[25]. The estimated genetic correlations were 1.00 (s.e. 0.04) between UKB and 23andMe, 0.84 (s.e. 0.05) between UKB and FinnGen, and 0.87 (s.e. 0.03) between 23andMe and FinnGen. The lower genetic correlation between FinnGen and the other two studies could be due to differences in the case definitions (triptan purchases in FinnGen vs. self-reporting in UKB and 23andMe). A comparable level of genetic correlation (0.81) has been reported before between primary care and self-reported migraine cases within UKB[26]. Another source of possible heterogeneity in effect sizes is the difference in genetic ancestry (Finnish in FinnGen vs. Non-Finnish European in the other two).

The genomic inflation factor ($\lambda_{GC}$) of the migraine meta-analysis was 1.38. There was a linear relationship between the association statistic and the LD-score (Supplementary Fig. 1), indicating that the polygenic background of migraine was the main source of the genomic inflation. However, as the intercept from LDSC[27] was elevated to 1.09 (s.e. 0.01) from its null value of 1.0, some inflation could also be due to confounding factors such as cryptic relatedness, population stratification, or other model misspecification, which we will take into account when reporting new loci. We further checked the LDSC intercepts for the individual studies: 1.03 (s.e. 0.01) for 23andMe, 1.00 (s.e. 0.01) for UKB and 1.10 (s.e. 0.01) for FinnGen. The higher intercept for FinnGen could be due to a different GWAS analysis method (whole genome-regression by REGENIE[28], including related samples) compared to UKB and 23andMe (logistic regression excluding related samples). Estimated SNP-heritability was 11.49% (s.e. 0.47%) from LDSC when population prevalence was assumed to be 16%.

We followed the locus definition of Hautakangas et al. (2022) and defined the LD-independent genome-wide significant (GWS; $P < 5 \times 10^{-8}$) risk loci from the meta-analysis iteratively by choosing the variant with the smallest $P$-value as an index variant and excluding all other GWS variants with LD $r^2 > 0.1$ to that index variant from further considerations until no GWS variants remained. Next, we formed a high LD region around each index variant extending to the level of $r^2 > 0.6$, and merged regions that were closer than 250 kb. Lastly, all other GWS variants were included in their closest region, and the region boundaries were updated, and once again regions closer than 250 kb were merged (see further details in Methods). Based on this locus definition, we identified 122 LD-independent risk loci. Of these, 35 were new

**Table 1 | Three study collections are included in the migraine meta-analysis**

| Study | Ancestry | cases | controls | N | Case % | Migraine definition | LD availability |
|---|---|---|---|---|---|---|---|
| UK Biobank | European, British | 10,881 | 330,169 | 341,050 | 3% | Self-reported | In-sample |
| 23andMe, Inc | European descent | 53,109 | 230,876 | 283,985 | 19% | Self-reported | In-sample for 26/102 fine-map regions |
| FinnGen R8 | European, Finnish | 34,385 | 308,114 | 342,499 | 10% | Medication purchases | In-sample |
| Meta-analysis | European descent | 98,374 | 869,160 | 967,534 | 10% | Self-reported medication purchases | In-sample 26/102, reference LD 76/102 |

(Table 2), defined by no previously reported migraine risk variant[4–14] residing within 250 kb from the locus boundaries (Fig. 1, Supplementary Data 1, Supplementary Figs. 2–4). Of the new loci, *TUBG2* has been implicated in a transcriptome-wide association study on migraine[29] and *ELAVL2* in a joint analysis of depression and migraine[30]. After genomic control by the LDSC intercept, 22 novel loci remained GWS (Table 2).

We observed statistically significant heterogeneity ($P < 0.05/122$) in effect sizes between the study collections only for two lead variants, both of which resided in the previously known migraine loci *PRDM16* and near *ZCCHC14* (Supplementary Data 1, Supplementary Fig. 3). As external replication data of 34,807 cases and 193,475 controls, we meta-analyzed data from the Trøndelag Health Study (HUNT)[31] and IHGC16 migraine meta-analysis excluding the Finnish cohorts and the 23andMe data[9]. Of the 35 lead variants of our new loci, 32 were consistent in direction ($P = 2.1 \times 10^{-7}$, one-sided binomial test), 17 replicated with $P < 0.05$ (one-sided test; Supplementary Data 2), and the *IPO8* locus was validated at $P < 0.05/35$ in the replication data. When we meta-analyzed the discovery and the replication data at the lead variants, 28 out of the 35 novel loci remained GWS (Supplementary Data 2).

To define the fine-map regions, we merged the risk loci that were closer than 1.5 Mb. This resulted in 102 fine-map regions. To avoid problems due to varying sample sizes across the variants, we included in fine-mapping only autosomal SNPs that were available in all three cohorts. This criterion reduced the number of common variants (MAF > 0.05) per regions on average by 19%.

## Comparison of different LD panels in fine-mapping

A common problem in meta-analyses is that the in-sample LD is not available, and the use of reference LD may lead to biased results. Figure 2 demonstrates this problem at the locus around *TSPAN2* where the optimal fine-mapping using the in-sample LD disagrees strongly with the UKB reference LD but agrees well with a more accurate UK biobank + FinnGen (UKB-FG) reference LD. This shows that, in our setting, fine-mapping based on the UKB-FG reference LD has the potential to yield reliable results, but that we need some way to assess, for each region, whether the reference LD has provided reliable results. Therefore, we evaluated whether some statistics, either derived from the GWAS results or from the fine-mapping results, could flag the regions where the reference LD produced unreliable fine-mapping results compared to the in-sample LD. The FINEMAP software outputs a posterior expectation of the number of causal variants (PENC), that is, an estimate of the number of independent causal variants in the region fine-mapped. Previously, PENC has been used to filter FINEMAP results in the schizophrenia fine-mapping study[24], and we chose it as one of our candidate statistics. Additionally, we considered three statistics derived from the set of top variants of the credible sets: maximum pairwise $r^2$, maximum marginal *P*-value, and minimum INFO value. The fifth statistic was a general measure of consistency between the LD matrix and GWAS results implemented in the susieR software[32]. We used the maximum difference of the variant-specific posterior inclusion probabilities (maxΔ) between the reference LD and the in-sample LD to assess the quality of the reference LD results in the 26 regions where the in-sample LD was available. A small maxΔ value (close to 0) indicates high quality (the reference LD produces similar results to the in-sample LD), and a large value (close to 1) indicates low quality (the reference LD produces different results from the in-sample LD).

In general, both LD reference panels performed well in most of the 26 regions available for this comparison, but, as expected[22], the more accurate UKB-FG panel performed clearly better than the UKB panel alone. For example, maxΔ was above 0.1 only in 2/26 regions with the UKB-FG panel but in 8/26 regions with the UKB panel (Fig. 3a).

We then investigated how well the five statistics could separate the regions with low-quality fine-mapping results from those with high-quality results for the UKB-FG reference LD. We observed that the PENC measure was able to retain the largest number of high-quality regions at the threshold where the low-quality regions were filtered out (Supplementary Figs. 5, 6). The reason why a high value of PENC can indicate possible problems with fine-mapping is because a mismatch between the reference LD and GWAS results often leads to several spurious signals in fine-mapping.

To define suitable thresholds for PENC, we observed that all low-quality regions (defined as maxΔ > 0.1) had PENC > 3 with the UKB-FG panel (Fig. 3b). Thus, we expect that this threshold has a high sensitivity to filter out the low-quality results in our analyses. The exact value of a reliable PENC threshold depends on the accuracy of the LD panel. For example, in our case, the less accurate UKB LD panel would require a more stringent threshold of PENC > 2 (Fig. 3b). Previously, a PENC value of 3.5 has been used in a schizophrenia GWAS fine-mapping with a high-quality LD panel[24]. Next, we evaluated how PENC classifies the 76 fine-map regions where only the reference LD was available. The 76 gray points in Fig. 3b show that the fine-map regions without the in-sample LD typically have PENC < 2.5 and only 6 of the 76 regions have PENC > 3 with the UKB-FG LD. Thus, while filtering by the PENC threshold of 3 will potentially exclude the regions that truly have more than three detectable causal variants, the number of such loci remains at most 6 in our analysis.

## FINEMAP results overview

Overall, for a majority of the fine-map regions, FINEMAP suggested one (42%) or two (46%) causal variants (Supplementary Data 3, Fig. 4.). The 102 fine-map regions together had 181 distinct signals when the signals were defined by the number of causal variants per region with the highest posterior probability. Among the 76 regions without the in-sample LD, 6 had PENC above 3. We flagged these regions to be of low quality, and their interpretation requires extra caution. The largest PENC observed was 5 and it occurred for two fine-map regions: *PRDM16* (index variant rs10218452) and *HOXB3* (index variant rs2555111). Of these, *HOXB3* region is flagged as low quality because there is no in-sample LD available.

The sizes of 95% credible sets ranged from 1 to 2787 variants, and 49 credible sets had 10 variants or less. A very high PIP ($\geq 0.9$) was observed for 10 variants (Supplementary Data 4), of which seven were in the high-quality fine-map regions (Table 3). We conducted a look-up from the Variant Effect Predictor (VEP) database for all credible sets to search for variants that are predicted to have functional consequences. In total, 149 unique missense variants were found of which 3 had PIP > 0.5: rs6330 (PIP = 0.59) in *NGF* located at chromosome 1, rs1133400 (PIP = 0.93) in *INPP5A* located at chromosome 10 and rs28929474 (PIP = 0.64) in *SERPINA1* located in a low-quality fine-map region at chromosome 14 (Table 3, Supplementary Data 5). Of these, rs6330 is a significant *cis*-eQTL for *NGF-AS1* expressed in atrial appendage of heart and rs28929474 for IFI27L2 expressed in tibial artery and in left ventricle of heart in GTEx v.08 data.

*NGF* encodes protein nerve growth factor beta (NGHβ) that is important in the development and survival of neurons, and involved in transmission of pain, temperature, and touch sensations via sensory neurons. It binds to two receptors, NTRK1 encoded by *NTRK1* and NGFR/p75NTR encoded by *NGFR*. Of note, two additional missense variants among the credible sets, rs6339 (PIP = 0.48) and rs6336 (PIP = 0.39), are located in *NTRK1* in a separate locus. The missense variant rs6330 shows association with multiple diseases of the musculoskeletal system and connective tissue including spinal stenosis, spondylosis, spondylopathies and hallux valgus in FinnGen R10 PheWAS scan, all to the opposite direction compared to the migraine risk (Supplementary Data 6).

*INPP5A* encodes a membrane-associated type I inositol 1,4,5-trisphosphate 5-phosphate protein, which hydrolyzes Ins(1,4,5)P3 leading to the mobilization of intracellular calcium. It has a central role in

**Table 2 | New 35 migraine risk loci identified from the inverse-variance weighted fixed-effects meta-analysis of 98,374 migraine cases and 869,160 controls**

| Locus name | Chromo-some | RSID | Position GRCh37 | Effect allele | Other allele | Effect allele frequency | Odds ratio | 95%CI | P-value |
|---|---|---|---|---|---|---|---|---|---|
| near RUNX3 | 1 | rs71014329 | 25348950 | I | D | 0.604 | 1.03 | (1.02-1.05) | 2.57E-10 |
| ST3GAL3 | 1 | rs783302 | 44366341 | G | A | 0.878 | 1.05 | (1.03-1.06) | 1.68E-09 |
| SF3B4 | 1 | rs7544531 | 149897217 | T | C | 0.084 | 1.08 | (1.05-1.10) | 5.08E-09 |
| near DTL^ | 1 | rs61830764 | 212289976 | A | G | 0.382 | 1.03 | (1.02-1.04) | 3.71E-08 |
| near APLF | 2 | rs112706954 | 68819969 | G | A | 0.023 | 1.15 | (1.11-1.19) | 7.88E-16 |
| TMEM131* | 2 | rs2305142 | 98375722 | G | A | 0.322 | 1.03 | (1.02-1.04) | 1.18E-08 |
| near GPD2^ | 2 | rs74482068 | 157560108 | D | I | 0.039 | 1.08 | (1.05-1.11) | 1.76E-08 |
| near RANP7* | 3 | rs11386839 | 22929430 | D | I | 0.5 | 1.03 | (1.02-1.04) | 7.68E-09 |
| ADD1 | 4 | rs10026792 | 2862190 | G | A | 0.687 | 1.03 | (1.02-1.04) | 2.79E-09 |
| EPHA5 | 4 | rs147908403 | 66362482 | C | T | 0.054 | 1.07 | (1.05-1.10) | 2.80E-09 |
| ITGA1^ | 5 | rs4865540 | 52184268 | C | A | 0.82 | 1.04 | (1.02-1.05) | 1.41E-08 |
| near GLRA1 | 5 | rs372257780 | 151200938 | I | D | 0.599 | 1.03 | (1.02-1.05) | 2.27E-09 |
| KCNIP1^ | 5 | rs78151838 | 170108683 | A | G | 0.905 | 1.06 | (1.04-1.08) | 1.82E-08 |
| MAML1^ | 5 | rs10794701 | 179181061 | A | G | 0.119 | 1.04 | (1.03-1.06) | 3.57E-08 |
| near COX19^ | 7 | rs117303395 | 1001963 | A | G | 0.019 | 1.13 | (1.08-1.18) | 4.40E-08 |
| MAD1L1 | 7 | rs10479762 | 2045351 | T | C | 0.419 | 1.03 | (1.02-1.04) | 8.01E-09 |
| ELAVL2*^ | 9 | rs10966033 | 23705736 | G | T | 0.617 | 1.03 | (1.02-1.04) | 2.70E-08 |
| near ZCCHC7 | 9 | rs10973207 | 37100525 | T | G | 0.187 | 1.04 | (1.03-1.06) | 1.04E-10 |
| near LMX1B | 9 | rs4358894 | 129464802 | C | G | 0.513 | 1.03 | (1.02-1.04) | 3.33E-09 |
| near DENND5A | 11 | rs34494849 | 9287030 | C | T | 0.768 | 1.03 | (1.02-1.05) | 1.17E-08 |
| near MTCH2 | 11 | rs11039324 | 47665686 | G | A | 0.601 | 1.03 | (1.02-1.04) | 9.76E-09 |
| MRE11A* | 11 | rs639311 | 94205747 | C | T | 0.681 | 1.03 | (1.02-1.04) | 9.02E-10 |
| IPO8 | 12 | rs12369125 | 30807195 | A | C | 0.251 | 1.04 | (1.02-1.05) | 7.08E-10 |
| MGAT4C | 12 | rs73187675 | 86409247 | T | A | 0.193 | 1.04 | (1.02-1.05) | 6.08E-09 |
| RP11-562L8.1*^ | 14 | rs1957110 | 29777492 | T | C | 0.409 | 1.03 | (1.02-1.04) | 1.59E-08 |
| INSM2 | 14 | rs2296919 | 36005659 | T | C | 0.807 | 1.04 | (1.03-1.05) | 3.44E-09 |
| RPS6KA5*^ | 14 | rs117151272 | 91415550 | A | T | 0.026 | 1.1 | (1.06-1.14) | 3.59E-08 |
| near ONECUT1^ | 15 | rs1899730 | 53166138 | T | G | 0.707 | 1.03 | (1.02-1.04) | 2.11E-08 |
| FAM174B | 15 | rs12910861 | 93218540 | C | T | 0.227 | 1.04 | (1.02-1.05) | 2.15E-09 |
| FAM65A | 16 | rs9934328 | 67573367 | C | G | 0.137 | 1.05 | (1.04-1.07) | 1.32E-11 |
| TUBG2 | 17 | rs2292750 | 40811781 | C | T | 0.452 | 1.03 | (1.02-1.04) | 3.53E-09 |
| near NRTN^ | 19 | rs76899991 | 5822370 | G | T | 0.963 | 1.08 | (1.05-1.11) | 2.89E-08 |
| SYMPK | 19 | rs74821481 | 46320041 | G | T | 0.678 | 1.04 | (1.03-1.05) | 4.59E-11 |
| near SERHL2^ | 22 | rs141478056 | 42939927 | G | A | 0.12 | 1.05 | (1.03-1.06) | 2.23E-08 |
| near FTHL17*^ | 23 | rs149675702 | 31063624 | C | T | 0.945 | 1.08 | (1.05-1.11) | 4.56E-08 |

RSID reference SNP ID, GRCh37 Genome Reference Consortium Human Build 37, 95% CI = 95% confidence interval. Alleles D and I refer to deletion and insertion, respectively. Odds ratio, 95%CI, and uncorrected two-sided $P$-value from the inverse-variance weighted fixed-effects migraine meta-analysis. *Uncorrected two-sided $P$-value > $5 \times 10^{-8}$ after genomic control by LD-score regression intercept. ^$P$-value > $5 \times 10^{-8}$ after data shown here were combined with replication data (Supplementary Data 2).

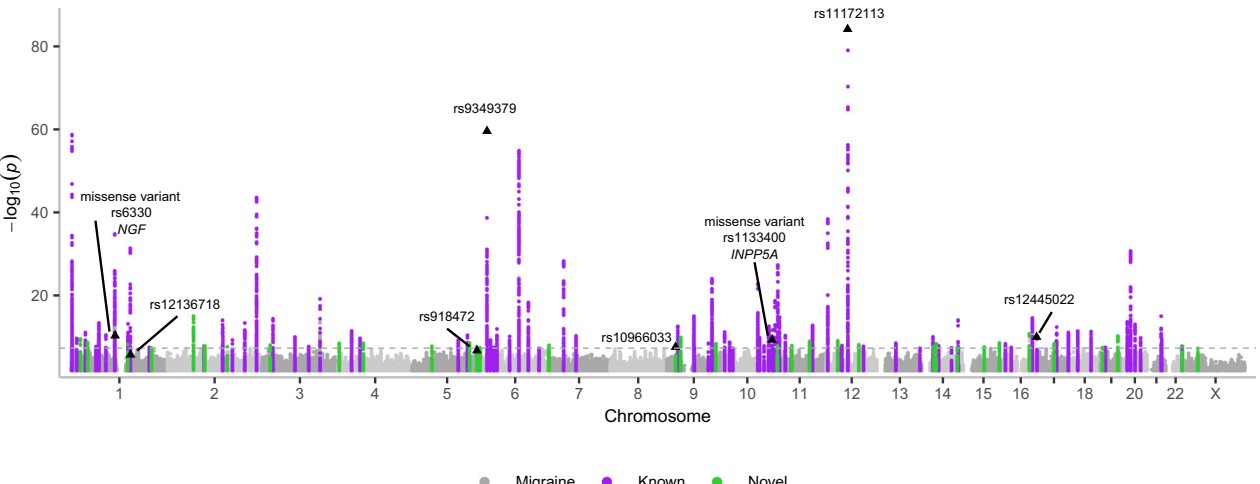

**Fig. 1 | A Manhattan plot of the inverse-variance weighted fixed effects migraine meta-analysis, including 98,374 cases and 869,160 controls.** X-axis presents the chromosomal location and y-axis -log$_{10}$ (uncorrected two-sided *P*-value). Known loci are highlighted in purple and new loci in green. Variants with posterior inclusion probability (PIP) > 0.9 and missense variants with PIP > 0.5 in high-quality fine-mapping regions are annotated. Plotted are variants with *P* < 0.01.

various cellular signaling processes including neurotransmission, hormone secretion, cell proliferation and muscle contraction. *INPP5A* is highly expressed in Purkinje cells of cerebellum, and in mice studies its deletion have been shown to cause ataxia and cerebellar degeneration[33,34].

*SERPINA1* encodes an alpha-1 antitrypsin, a serine protease inhibitor protein, that belongs to the serpin superfamily. Its primary target is elastase, and other targets are plasmin and thrombin. Several mutations, including our high-PIP variant rs28929474C>T, in *SERPINA1* can cause an autosomal co-dominant genetic disorder alpha-1 antitrypsin (AAT) deficiency, which can lead to lung or liver disease due to reduced alpha-1 antitrypsin levels[35]. The missense variant rs28929474 is highly pleiotropic showing associations to 132 proteins (Supplementary Data 7) and multiple disease categories in PheWAS of FinnGen R10 data including, for example, diseases of the respiratory system, diseases of the circulatory system, diseases of digestive system, pregnancy related diseases, diseases of the nervous system, and diseases of musculoskeletal system and connective tissue (Supplementary Data 6,8,9).

Five additional high-impact variants on protein function were among the credible sets (stop-gained rs5758511 in *CENPM*, start-lost rs3825080 in *ARHGAP9*, and rs798488 in *GNA12* and splice-acceptor rs41298712 in *ENDOV* and rs9906358 in *TSPAN10*), but only with modest PIPs below 0.01 (Supplementary Data 5), and another 5 variants with high-impact on something else than protein coding function (long non-coding RNA, antisense or nonsense mediated decay) with PIPs below 0.02. Five variants with PIP above 0.1 were reported to have significant plasma protein associations by The Pharma Proteomics Project[36] (Supplementary Data 7).

Our results added information on two of the strongest known migraine risk loci by estimating PIPs of 1.00 for the intronic variants rs9349379 in *PHACTR1* and rs11172113 in *LRP1*, both of which have been previously prioritized[12]. We were able to fine-map both of these loci by using the in-sample LD. The candidate variant in *PHACTR1* is also associated with many vascular diseases and its effects on gene expression of the genes in the locus have been studied in detail but with contradicting results[37,38]. Also, the candidate variant in *LRP1* is associated with two proteins (Supplementary Data 7) and several vascular diseases, such as sporadic thoracic aortic dissection, fibromuscular dysplasia and spontaneous coronary artery dissection[39–41]. The LDL receptor-related protein 1 (LRP1) is a cell surface receptor and has an important role in vascular and blood-brain barrier integrity[42–44]. It is expressed in almost every tissue, and most studied in the liver and

brain. LRP1 is also involved in vascular calcium signaling by regulating smooth muscle cell contractility[43]. A recent study suggested that *LRP1* expression is regulated by an allele-specific mechanism of intronic rs11172113 located in an enhancer region through two transcription factors (MECP2 and SNAIL)[45].

Due to the restriction of including in fine-mapping only the variants that are available in all three data sets, the original lead variant was missing in 17/102 fine-map regions (Supplementary Data 3b). In 14/17 of these regions, the original lead variant was represented by one of the top credible set variants (defined as being in LD with $r^2 > 0.1$ in the UKB data). For the remaining 3 regions, the signal related to the original lead variant may be missing from the fine-mapping results, and we flagged these regions to be of low quality. Hence, when combined with the earlier 6 low-quality regions with PENC > 3, in total we reported 9 low-quality regions (Fig. 4). Among the fine-map regions for which the lead variant was included in the analysis, the lead variant was within the 95% credible sets in 83/85 fine-map regions and within the top configuration in 73/85 of the regions.

**Phenome-wide association scans for the credible set variants**

We conducted three separate phenome-wide association studies (PheWAS) by using data from FinnGen Data Freeze 10, including 429,209 individuals. First, by a PheWAS for the 181 credible set top variants and the list of 2399 FinnGen endpoints excluding the migraine endpoints, we identified 404 variant-disease associations with $P < 1 \times 10^{-5}$ (Supplementary Data 6, https://hhautakangas.github.io/phewas_migraine_tables.html). All associations remained significant at a false discovery rate (FDR) of 0.05. Of these, 108 variant-disease associations belonged to diseases of the circulatory system, including, for example, hypertension and ischemic heart disease, followed by 39 variant-trait associations in a category of quantitative endpoints, including, e.g., height and BMI, 34 in diseases of the musculoskeletal system and connective tissue category, including, e.g., spinal stenosis and rheumatoid arthritis, and 28 associations in diseases of the respiratory system, including, e.g., asthma and COPD.

Second, for the 159 functional variants among the credible sets, we conducted a targeted PheWAS scan within neurological and cardiovascular endpoints, and identified 122 variant-disease associations with $P < 1 \times 10^{-4}$ (Supplementary Data 8, https://hhautakangas.github.io/phewas_migraine_tables.html), including traits such as sleep apnea and stroke. All associations remained significant at an FDR of 0.05. Third, for the 307 variants with PIP > 0.1, with a similar targeted PheWAS scan within the neurological and cardiovascular endpoints, we

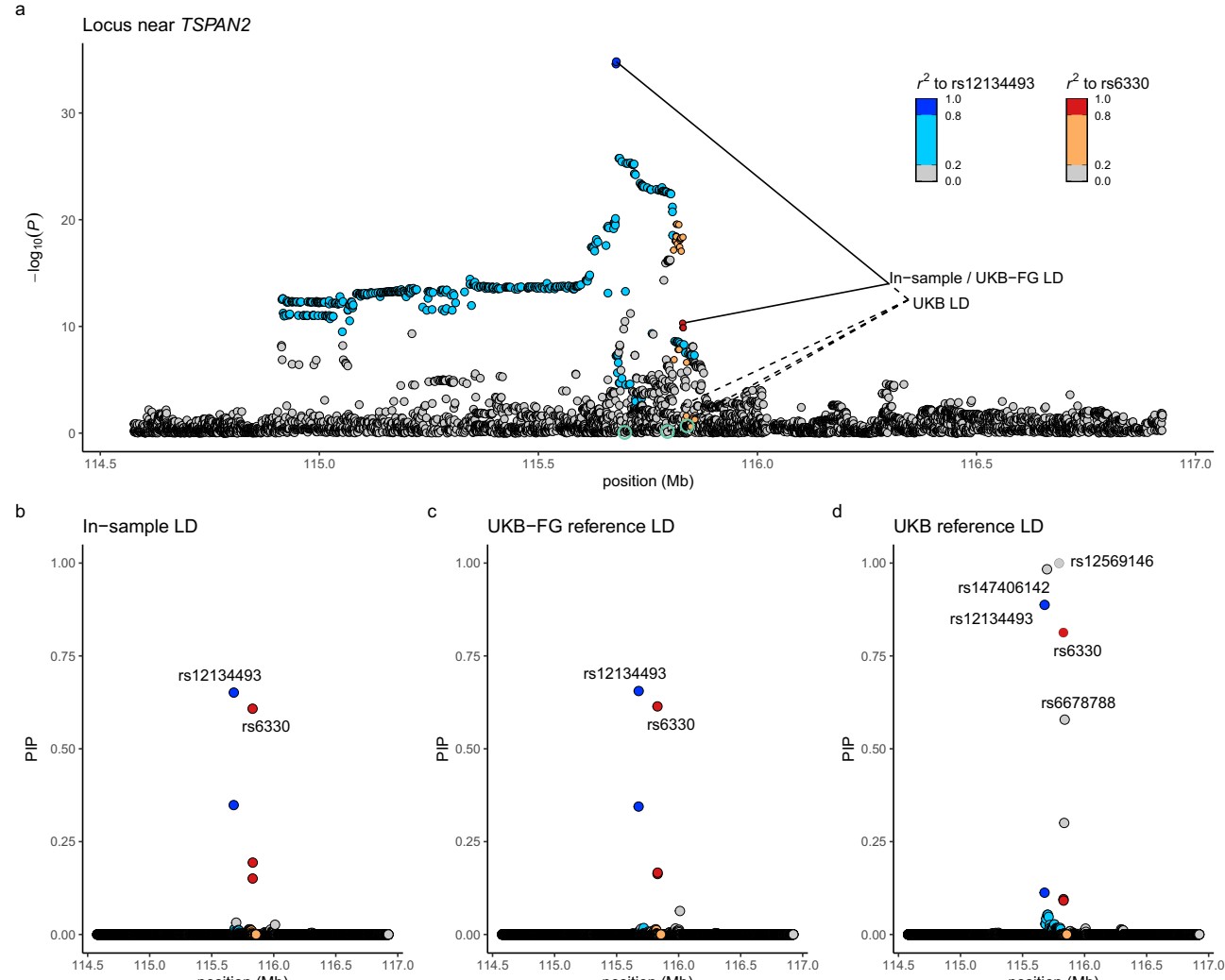

**Fig. 2 | Fine-mapping a region near TSPAN2 at chromosome 1 using three different LD sources. a** Plot of the GWAS results with the chromosomal location on x-axis and the strength of the association as -log₁₀ uncorrected two-sided *P*-values from the inverse-variance weighted fixed-effect meta-analysis with 98,374 migraine cases and 869,160 controls on y-axis. Variants are colored based on the squared correlation ($r^2$) to the two variants in the top configuration suggested by FINEMAP with the in-sample LD. The suggested top configurations based on three LD panels are marked by lines with the in-sample LD and the UK biobank + FinnGen (UKB-FG) reference LD giving the same top configuration and the UK biobank (UKB) reference LD including three additional variants (highlighted in green). Posterior inclusion probabilities (PIPs) for the variants based on **b** in-sample LD, **c** UKB-FG reference LD and **d** UKB reference LD.

identified 330 variant-disease associations with $P < 1 \times 10^{-4}$ (Supplementary Data 9, https://hhautakangas.github.io/phewas_migraine_tables.html), including, e.g., focal epilepsy and hydrocephalus. All associations remained significant at an FDR of 0.05.

## Discussion

Well over one hundred risk loci for migraine have been reported from GWAS, but the causal variants and genes are still mostly unknown[4–14]. Statistical fine-mapping of the GWAS results at the risk loci is a natural next step but reliable fine-mapping of large meta-analysis data has turned out to be very difficult. Our recent migraine meta-analysis of 25 studies[13] illustrated these difficulties as the accurate LD information was not available, and the sample size varied considerably across variants. In this study, our goal was to provide reliable fine-mapping for migraine by creating a new migraine meta-analysis for which accurate LD information was available, and the sample size across variants was more stable. Despite the more stringent selection criteria, the effective sample size of our new meta-analysis (339,000) turned out to be comparable to that of the earlier meta-analysis (326,000).

A key question in fine-mapping a GWAS meta-analysis is how to assess the reliability of the results. We were able to study this question by directly comparing results between accurate in-sample LD and approximate reference panel LD. We observed that the posterior expected number of causal variants (PENC), as reported by FINEMAP, distinguished well the regions with high-quality fine-mapping results from those with low-quality results. We also observed that an appropriate PENC threshold depends on the quality of the reference panel. In our case, we were able to use an upper limit of 3.0 for PENC. While this upper limit restricts our ability to fine-map the migraine risk regions that truly have more than 3 causal signals, we expect that the proportion of such regions is small, as only 3/26 (12%) of the migraine loci with the in-sample LD had PENC over 3 in our analysis.

Here, we performed the systematic fine-mapping of a migraine meta-analysis and provided high-quality fine-mapping results for 91% of the migraine risk regions identified by the meta-analysis. Our high-quality results highlight two missense variants with high PIPs: rs6330 (PIP = 0.59) in *NGF* and rs1133400 (PIP = 0.93) in *INPP5A*.

The variant rs6330 is only in weak LD ($r^2 = 0.04$) with the lead variant (rs12134493) of its locus and was identified as a secondary

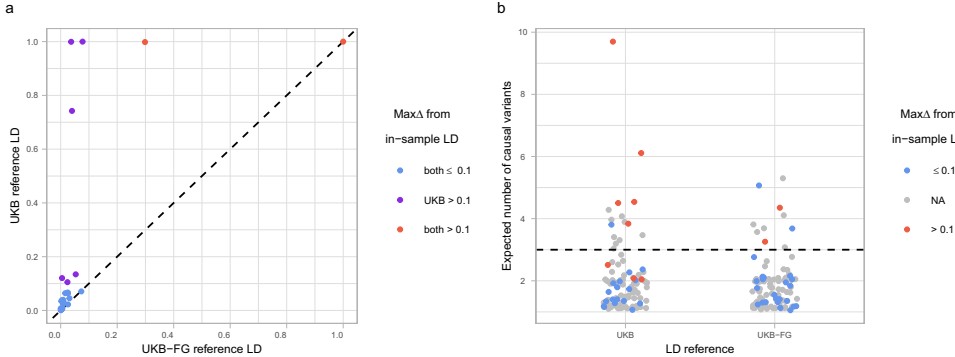

**Fig. 3 | Comparison of fine-mapping performance using different LD reference panels. a** Scatter plot comparing the maximum posterior inclusion probability (PIP) differences (maxΔ) between the in-sample and reference LD for 26 fine-map regions, where reference LD is based either on UK biobank + FinnGen (UKB-FG) or only on UK biobank (UKB). **b** Strip chart shows the posterior expected number of causal variants (PENC) from fine-mapping for the two LD reference panels for the 102 fine-map regions. Red dots indicate large differences from the in-sample LD (maxΔ > 0.1), and gray color indicates regions for which only reference LD is available and therefore maxΔ is not known. Horizontal line shows PENC = 3 that we use as a threshold to define reliable results with the UKB-FG panel.

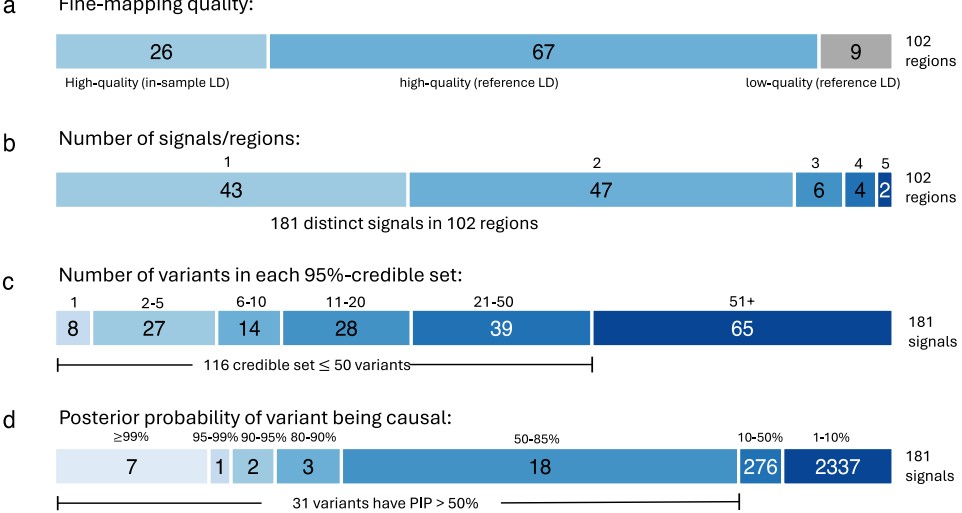

**Fig. 4 | Summary of the FINEMAP results across the 102 migraine risk regions. a** Quality of the fine-mapping with in-sample LD and reference LD from UK biobank + FinnGen. **b** Number of distinct signals for the fine-mapped regions. One signal at 43 loci and 2–5 signals at 59 loci. **c** Number of variants in 95% credible sets: eight signals were fine-mapped to a single variant. **d** Distribution of the posterior inclusion probabilities (PIP) of the variants in credible sets. Ten variants have PIP > 90%.

signal in our fine-mapping. A recent study[14] has also reported that the migraine association of rs6330 remained statistically significant in a conditional analysis after adjusting for the stronger signal (rs2078371) within the same risk locus. *NGF* has been reported to be highly expressed in hippocampus and cortex[46,47] although according to the GTEx v8 data, *NGF* does not show statistically significant expression in any brain tissue but shows high expression in multiple other tissues, including, for example, ovary, tibial nerve, arteries, visceral adipose, and heart. NGF levels have been reported to be elevated in cerebrospinal fluid in chronic migraine patients compared to controls[48], and decreased in blood serum of episodic migraine patients compared to controls and chronic migraine patients[49]. The potential causality of the pathway related to *NGF* was further supported by two additional missense variants, rs6339 (PIP = 0.48) and rs6336 (PIP = 0.39), located in *NTRK1* which encodes one of the two receptors for NGF. NGF and its receptors have a central role in the pain perception, and elevated NGF levels have been observed also in many other chronic pain conditions, such as osteoarthritis and low back pain[50–52]. Multiple antibodies of NGF or small molecular inhibitors of the NGF receptors have been developed and tested in clinical studies to treat chronic pain conditions, including low back pain and osteoarthritis[53–57]. Even though

some candidate drugs have shown potential benefit relating to pain relief, an increased risk of progressive osteoarthritis has been observed in a small group of the treated patients[57], and therefore, none of the drugs have yet received FDA approval. Currently, other types of drug classes (p75 neurotrophin receptor fusion protein, LEVI-04 (ClinicalTrials.gov ID: NCT05618782) and anti-NGF PEGylated Fab' antibody[58]), are being developed and in pre-clinical or clinical testing. In adults, after pain stimuli, NGF activates the overexpression of other neuronal molecules, including calcitonin gene-related peptide (CGRP) and substance P[57]. CGRP is involved in migraine pain, and several effective monoclonal antibodies targeting either CGRP or its receptors have been developed to treat migraine[59–61].

Gene *INPP5A* is highly expressed in Purkinje cells of cerebellum[62] and involved in multiple cellular signaling processes, including neurotransmission, hormone secretion, cell proliferation, and muscle contraction through its role in the pathway regulating intracellular calcium levels. The missense variant rs1133400 is in modest LD ($r^2 = 0.36$) with the lead variant of the locus (rs200314499) that was filtered out from the fine-mapping in the quality control phase. For this locus, FINEMAP suggested two causal variants (PENC = 1.65). PheWAS showed no other associations with this missense variant.

**Table 3 | Variants with high (>0.9) posterior inclusion probability (PIP) and missense variants with PIP > 0.5 among the 93 high-quality fine-map regions**

| Gene (VEP) | Predicted consequence (VEP) | RSID | Position GRCh37 | Chromosome | Effect allele | Other allele | PIP | Effect allele frequency | Odds ratio (95% confidence interval) | P-value | LDsource |
|---|---|---|---|---|---|---|---|---|---|---|---|
| PHACTR1 | Intron variant | rs9349379 | 12903957 | 6 | A | G | 1.000 | 0.578 | 1.09 (1.08-1.10) | 2.59E-60 | in-sample |
| LRP1 | Intron variant | rs11172113 | 57527283 | 12 | T | C | 1.000 | 0.596 | 1.11 (1.10-1.12) | 7.27E-85 | in-sample |
| - | Intergenic variant | rs12445022 | 87575332 | 16 | G | A | 1.000 | 0.667 | 1.04 (1.03-1.05) | 1.04E-10 | in-sample |
| - | Intergenic variant | rs12136718 | 156409585 | 1 | A | G | 0.999 | 0.072 | 1.05 (1.03-1.07) | 1.95E-06 | in-sample |
| ELAVL2 | Intron variant | rs10966033 | 23705736 | 9 | G | T | 0.954 | 0.617 | 1.03 (1.02-1.04) | 2.70E-08 | UKB-FG |
| TLX3 | 3' UTR variant | rs918472 | 170738836 | 5 | A | G | 0.932 | 0.708 | 1.03 (1.02-1.04) | 1.95E-07 | UKB-FG |
| INPP5A | Missense variant | rs1133400 | 134459388 | 10 | G | A | 0.926 | 0.198 | 1.04 (1.03-1.05) | 5.06E-10 | UKB-FG |
| NGF | Missense variant | rs6330 | 115829313 | 1 | A | G | 0.593 | 0.461 | 1.03 (1.02-1.04) | 4.97E-11 | in-sample |

RSID = reference SNP ID, GRCh37 = Genome Reference Consortium Human Build 37, Odds ratio, 95% confidence interval, and uncorrected two-sided P-value obtained from the inverse-variance weighted fixed-effects migraine meta-analysis. PIP obtained from fine-mapping by FINEMAP.

Another important finding is in the *PHACTR1* locus, which is one of the strongest known migraine risk loci. There, our fine-mapping suggested one causal variant (PENC = 1.29), with the lead variant rs9349379 being a clear candidate for being causal with PIP of 1.00, consistent with previous results[12]. In our FinnGen PheWAS, we also detected strong associations between the variant and, for example, major coronary disease events ($P = 8.22 \times 10^{-52}$), ischemic heart disease ($P = 1.18 \times 10^{-38}$), and angina pectoris ($P = 7.71 \times 10^{-26}$), all in the opposite direction compared to migraine risk. Because of these well-known associations with multiple vascular diseases, this locus has been previously studied in detail, but with contradicting results. Gupta et al. (2017)[37] reported that rs9349379 regulates upstream gene *EDN1*, whereas Wang et al. (2018)[38] reported that they failed to replicate this endothelial rs9349379-*EDN1* eQTL, but instead showed that rs9349379 regulates the closest gene *PHACTR1*, confirming previous vascular rs9349379-*PHACTR1* eQTLs. Further, Rubin et al (2022)[63] observed that a loss of *PHACTR1* gene does not seem to have any effect on the endothelial or smooth muscle cells of the transgenic mice, and suggested that *PHACTR1* has no contribution to pathological vascular phenotype in mice through cells involved in vascular physiology. Our fine-mapping has provided strong evidence that the lead variant rs9349379 is causal for migraine, but given that the variant is intronic, our fine-mapping results alone do not provide direct evidence through which gene or mechanism this association affects the disease risk.

Our study has some limitations. First, since reliable fine-mapping requires that we exclude variants that are not present in all three component studies of our meta-analysis, it is possible that we also exclude some of the true causal variants. This is a potential problem especially when some of the top variants of the fine-map region have been filtered out from fine-mapping. To identify the regions that are likely to be affected by this problem, we studied the LD patterns between the fine-mapped variants and the top variants from the fine-map regions that were not included in the fine-mapping analysis. For most (14/17) regions where the top variants were missing from fine-mapping, the signal of the top variant was at least partly represented by another variant in LD with the top variant. Additionally, since very rare variants or multi-allelic variants were not included in our analysis, we miss the true causal variants that are rare or multi-allelic. Since our variant set is not comprehensive, we must keep in mind that also variants that have a very high probability of being causal in our analysis may still have such variants in high LD that were not included in our analysis. A valid calibration of the PIPs would require that all potential causal variants be included in the analysis. In practice, for common variants, this would require comprehensively imputed data sets with no missing variants in any of the meta-analyzed studies, and, for rare variants, the availability of high coverage sequencing data. Currently, we do not yet have such resources available in typical GWAS meta-analyses of common diseases such as migraine. Summary statistics imputation methods[64,65] could, in principle, complete the GWAS results in the individual data sets before the meta-analysis, but as the imputation would use reference LD rather than the in-sample LD, reliability of the subsequent fine-mapping results would still remain unclear.

Another limitation of our study relates to the phenotype definitions of different substudies. First, both the UKB and 23andMe GWAS are based on self-reported migraine status, and therefore, some other conditions, such as tension headache, may have been wrongly reported as migraine for some cases. Second, the FinnGen GWAS is based on triptan purchase data, which may represent a specific subset of migraine patients. Triptans are not suitable for all migraineurs and, especially, they are contraindicated in patients with cardiovascular diseases. Overrepresentation of migraineurs without any cardiovascular diseases could lead to some FinnGen PheWAS associations where migraine risk alleles seem to have a protective effect on cardiovascular phenotypes, while observational studies have reported that both

migraine and cardiovascular disease risks in women are positively associated[66]. We observed that the genetic correlation between Finn-Gen and the other two data sets was slightly below 1.0 but we found little evidence for heterogeneity in the effect size estimates at the reported risk loci between the data sets. In future, methods that explicitly model both the heterogeneous effect sizes and LD patterns across the data sets may improve the locus discovery in GWAS meta-analyses[67].

To conclude, we performed a migraine GWAS meta-analysis with 98,375 migraine cases and 869,159 controls and identified 122 risk loci of which 35 were new. We followed up the meta-analysis by the systematic fine-mapping analysis of migraine risk loci and identified 7 variants with a high probability of being causal. In addition to providing new information about genetic risk of migraine, we also proposed how one could, in general, evaluate whether the fine-mapping results of each risk loci seem reliable based only on the output from the fine-mapping software FINEMAP. While a definitive fine-mapping analyses will require more comprehensive data than are currently available for the GWAS meta-analyses of common diseases, our study shows how reliable and novel fine-mapping results can be extracted already from the currently available data sets by a suitable analysis approach.

## Methods

### Data

We performed a new migraine meta-analysis by combining summary statistics from three migraine GWAS: UK Biobank ($N = 341,050$, 10,881 cases and 330,169 controls), 23andMe ($N = 283,985$, 53,109 cases and 230,876 controls), and FinnGen R8 ($N = 342,499$, 34,385 cases and 308,114 controls). By meta-analyzing the three studies, the total sample size was 967,534, including 98,375 migraine cases and 869,159 controls.

UK Biobank: The UK Biobank project is a population-based prospective cohort study that consists of over 500,000 participants aged 40-69 at recruitment collected from several regions across the United Kingdom. The participants completed questionnaires, attended interviews, and clinical examinations by a trained staff member. A detailed description of UK Biobank is provided elsewhere[68], and detailed genotyping, quality control and imputation procedures are described at the UK Biobank website (https://www.ukbiobank.ac.uk/). We used the migraine GWAS data described in ref. 13 with self-reported migraine as the phenotype. UK Biobank received ethical approval from the North West Multi-centre Research Ethics Committee (MREC) and informed consent has been obtained from all participants.

23andMe: 23andMe migraine GWAS was performed by a personal genomics company 23andMe, Inc. (https://www.23andme.com/) and detailed description of the migraine GWAS is provided elsewhere[8]. All participants have provided informed consent and filled an online survey according to 23andMe's human subjects protocol, which was reviewed and approved by Ethical & Independent Review Services, a private institutional review board. Briefly, migraine cases were assessed from the participants that had reported migraine or answered "Yes" to any of the questions related to migraine, and controls from participants that did not report having migraine or answered "No" to all of the questions related to migraine, excluding participants with discordant answers.

FinnGen: FinnGen (https://www.finngen.fi/en) is a large biobank study that has collected and genotyped 500,000 Finns and combined these data with longitudinal registry data including The National Hospital Discharge Registry, Causes of Death Registry and medication reimbursement registries, all of these linked by unique national personal identification codes. FinnGen includes prospective and retrospective epidemiological and disease-based cohorts and hospital biobank samples. A detailed description of FinnGen is provided in ref. 69. We used the 8th Data Freeze for the migraine GWAS. The migraine cases were defined as the individuals who had at least one triptan purchase and the remaining individuals without any triptan purchases were defined as controls from the social insurance institution of Finland (KELA) registry including medication reimbursement and drug purchases (https://r8.risteys.finngen.fi/phenocode/MIGRAINE_TRIPTAN).

FinnGen participants provided informed consent under the Finnish Biobank Act. Older cohorts with study-specific consents were transferred to the Finnish biobanks after approval by Fimea, the National Supervisory Authority for Welfare and Health. Recruitment protocols followed the biobank protocols approved by Fimea. The Coordinating Ethics Committee of the Hospital District of Helsinki and Uusimaa (HUS) approved the FinnGen study protocol (Nr HUS/990/2017).

The FinnGen study is approved by Finnish Institute for Health and Welfare (permit numbers: THL/2031/6.02.00/2017, THL/1101/5.05.00/2017, THL/341/6.02.00/2018, THL/2222/6.02.00/2018, THL/283/6.02.00/2019, THL/1721/5.05.00/2019 and THL/1524/5.05.00/2020), Digital and population data service agency (permit numbers: VRK43431/2017-3, VRK/6909/2018-3, VRK/4415/2019-3), the Social Insurance Institution (permit numbers: KELA 58/522/2017, KELA 131/522/2018, KELA 70/522/2019, KELA 98/522/2019, KELA 134/522/2019, KELA 138/522/2019, KELA 2/522/2020, KELA 16/522/2020), Findata permit numbers THL/2364/14.02/2020, THL/4055/14.06.00/2020,THL/3433/14.06.00/2020, THL/4432/14.06/2020, THL/5189/14.06/2020, THL/5894/14.06.00/2020, THL/6619/14.06.00/2020, THL/209/14.06.00/2021, THL/688/14.06.00/2021, THL/1284/14.06.00/2021, THL/1965/14.06.00/2021, THL/5546/14.02.00/2020, THL/2658/14.06.00/2021, THL/4235/14.06.00/2021 and Statistics Finland (permit numbers: TK-53-1041-17 and TK/143/07.03.00/2020 (earlier TK-53-90-20) TK/1735/07.03.00/2021).

The Biobank Access Decisions for FinnGen samples and data utilized in FinnGen Data Freeze 8 include: THL Biobank BB2017_55, BB2017_111, BB2018_19, BB_2018_34, BB_2018_67, BB2018_71, BB2019_7, BB2019_8, BB2019_26, BB2020_1, Finnish Red Cross Blood Service Biobank 7.12.2017, Helsinki Biobank HUS/359/2017, Auria Biobank AB17-5154 and amendment #1 (August 17 2020), AB20-5926 and amendment #1 (April 23 2020), Biobank Borealis of Northern Finland_2017_1013, Biobank of Eastern Finland 1186/2018 and amendment 22 § /2020, Finnish Clinical Biobank Tampere MH0004 and amendments (21.02.2020 & 06.10.2020), Central Finland Biobank 1-2017, and Terveystalo Biobank STB 2018001.

We had access to the complete in-sample LD information for the UK Biobank and FinnGen samples via the individual-level genotype data. Additionally, we had access to the in-sample LD-matrices in 23andMe data for 26 of our fine-map regions. Thus, for the 26 fine-map regions, we were able to do a high-quality fine-mapping based on the in-sample LD while, for the remaining 76 regions, we needed to apply an LD reference panel that does not perfectly match the LD information corresponding to our GWAS summary statistics. To assess the effect of the LD reference panel, we formed two reference panels from the available LD information: one including data only from the UK Biobank (UKB), and the other combining the LD matrices from UK Biobank and FinnGen (UKB-FG), as explained in section "Fine-mapping".

### Statistical significance

The false positive rate was controlled based on the significance thresholds mentioned in the text; that is, the results were considered statistically significant when the corresponding $P$-value was below the significance threshold.

### Genetic association analyses

The UK Biobank and 23andMe GWAS had been conducted by logistic regression on migraine (using PLINK2[70] or custom software of the 23andMe Research Team, respectively), and the FinnGen GWAS by a whole-genome regression model for a binary trait with REGENIE[28].

All the samples were of European descent. Related individuals had been excluded by using a kinship value threshold of 0.0442 computed by KING[71] from UK Biobank, and by using a minimal expected amount of sharing between first cousins from a segmental identity-by-descent algorithm from 23andMe. For the FinnGen GWAS analysis, REGENIE accounted for the genetic relatedness by default, and therefore no relatedness exclusions were applied.

We excluded multi-allelic variants (0.2% in 23andMe, 0.5% in UK Biobank, and 0.8% in FinnGen), and variants with minor allele frequency (MAF) < 0.01, IMPUTE2 info or MACH $r^2$ < 0.6, and when available, missingness > 0.05 and Hardy-Weinberg equilibrium (HWE) $P < 1 \times 10^{-6}$ from each study. Consequently, we are only considering biallelic common variants in this work. We recoded indels as insertions (I) and deletions (D). We mapped the FinnGen GWAS summary statistics positions from hg38 to hg37 by UCSC LiftOver[72] (downloaded January 2021). We excluded the SNPs with an effect allele frequency (EAF) discrepancy of >0.30 and indels with an EAF discrepancy of >0.20 compared to UK Biobank from each study following Hautakangas et al. 2022.

We conducted an inverse-variance weighted fixed-effects meta-analysis to combine the three studies by GWAMA v2.1[73] with 11,316,120 variants, of which 7,062,924 variants were available in all three studies.

## Genetic correlation and SNP-heritability using LD Score regression

We estimated genetic correlations between the three GWAS and SNP-heritability from the migraine meta-analysis by LD Score regression v1.0.0[25,27] with precomputed 1000 Genomes European LD Scores (https://data.broadinstitute.org/alkesgroup/LDSCORE/) limiting the analysis to the HapMap3 SNPs. We used munge-tool to reformat and perform additional quality control for all GWAS summary statistics prior to the genetic correlation estimation. We obtained a liability scale SNP-heritability estimate[74] by using a population prevalence of 16% for migraine. The intercept of the LD score regression analysis can be used as a genomic control parameter to provide a conservative $P$-value by dividing the observed chi-square statistic of the GWAS association by the intercept before computing the $P$-value from the chi-square distribution[27]. We used this approach to indicate which variants in Table 2 had $P > 5 \times 10^{-8}$ after the genomic control.

## Locus definition

We followed the locus definition of Hautakangas et al. (2022) and defined an LD-independent genome-wide significant (GWS, $P < 5 \times 10^{-8}$) risk locus from the meta-analysis by using the UKB LD. Iteratively, we chose the variant with the smallest $P$-value as the index variant and excluded all variants that had $r^2 \geq 0.1$ with the index variant, until no variant had $P < 5 \times 10^{-8}$. Next, we formed high LD regions around each index variant based on the combined UKB-FG LD and $r^2$ threshold of 0.6. The start of the high LD region was the smallest position, and the end of the region was the largest position, where any variant had $r^2 > 0.6$ with the index variant. Next, we formed the loci by adding ± 250 kb around the high LD region and merged the overlapping regions. Further, we iteratively added all other GWS variants to their closest loci, and updated the loci boundaries if any of the variants added were outside the existing locus boundaries. Again, the overlapping loci were merged. We named each locus by the lead variant, i.e., the variant with the smallest $P$-value of the locus.

## Replication in HUNT All-in Headache and IHGC16

To replicate our new loci, we used two independent data sets with no overlaps with our GWAS data: HUNT All-in Headache[31] (N = 40,224, 7801 cases, 32,423 controls) and IHGC16 migraine meta-analysis[9] excluding 23andMe and the Finnish cohorts (N = 189,000, 27,006 migraine cases and 161,994 controls). The meta-analysis of the replication data thus contained N = 229,224 samples (34,807 cases and 194,417 controls). We used a one-sided $P$-value threshold of 0.05 to denote a replication and assessed consistency of the effect directions by a sign test. We also reported the two-sided $P$-value of a combined analysis of our discovery and replication results to determine which of the new loci remained GWS after observing the replication data.

## Fine-mapping

For fine-mapping, we first merged loci that were closer than 1.5 Mb, leading to 102 fine-map regions. We performed fine-mapping for each fine-map region with FINEMAP v1.4[19,22]. FINEMAP is a Bayesian method that uses summary statistics from a GWAS together with LD information to infer which variants are most likely causal within the genomic region. We used the default prior parameters and set the maximum number of causal variants to 10.

We estimated the in-sample LD correlations for the individual GWAS cohorts by using LDStore2 v2.0[22]. We combined the in-sample LD correlations for the meta-analysis data set by combining the study-specific LD matrices by weighting each matrix in proportion to its effective sample size as follows:

$$\mathbf{R} = (M_1\mathbf{R}_1 + \ldots + M_C\mathbf{R}_C)/M, \qquad (F1)$$

where $\mathbf{R}_i$ is the LD correlation matrix of study i, $M_i = 4N_i\,p_i\,(1\text{-}p_i)$ is the effective sample size of study i, with $N_i$ being the total sample size (i.e., the sum of cases and controls) and $p_i$ being the proportion of cases in study i, and $M = M_1 + \ldots + M_C$ is the sum of the effective sample sizes.

For the UK Biobank reference LD (UKB-LD), we used the in-sample LD estimated from the individuals included in the UKB GWAS.

For the combined UKB-FG LD reference panel, we combined the UKB and FG in-sample LD matrices by weighting FG in proportion to its effective sample size, and UKB in proportion to the combined UKB +23andMe effective sample size using the above formula (F1).

## LD reference panel sensitivity analyses

We compared the performance of different LD reference panels (UKB LD, UKB-FG LD and in-sample LD) on the FINEMAP results for the 26 fine-map regions for which the in-sample LD was available. We used the maximum difference between the posterior inclusion probabilities (PIPs) from different panels (maxΔ) to compare the performance of the three LD panels.

In addition, we examined five candidate statistics that could be used for separating the fine-map regions for which fine-mapping with the reference LD performs poorly when compared to the use of the in-sample LD. The first statistic was the posterior expectation of the number of causal variants (PENC) as outputted by FINEMAP. Next three statistics were defined from the top variant(s) of the credible set(s) determined by FINEMAP using the maximum pairwise $r^2$, the maximum marginal $P$-value from the meta-analysis, and the minimum INFO value. The fifth statistic was the regularization parameter ‘s’ of Zou et al.[32] that reflects the overall consistency between the LD-matrix and the GWAS z-scores and was computed using the function estimate_s_rss() from the R package susieR.

## Variant annotation and eQTL and pQTL mapping

FINEMAP reports 95% credible sets (CS). We searched for coding variants among the CS from the Ensembl Variant Effect Predictor (VEP) (http://grch37.ensembl.org/Homo_sapiens/Tools/VEP) database by using a default of 5 kb window around the index variant.

For the follow-up analyses, we formed a functional variant group among the CS variants by including the variants that were predicted by VEP to have a moderate or high impact on the transcript (https://www.ensembl.org/info/genome/variation/prediction/predicted_data.html). This includes transcript ablation, splice acceptor or donor variants, stop gained, frameshift variant, stop lost, start lost, transcript amplification, inframe insertion or deletion, and missense variant.

To assess variants' effects on gene expression, we used the list of significant eQTLs of the 49 tissues as reported by GTEx v.8 (https://gtexportal.org/home/). For assessing variants' effects on the plasma proteome, we used the list of significant pQTLs as reported by the UK biobank Pharma Proteomics Project in their Supplementary Table 10 and extracted the data from their website (https://metabolomips.org/ukbbpgwas/).

## Phenome-wide association scans

We performed three phenome-wide association scans (PheWAS). First, we scanned all 181 candidate variants of the risk loci (top variants of the credible sets) for associations with 2399 FinnGen Data Freeze 10 (R10) GWAS endpoints (excluding 9 migraine endpoints) at significance level $1 \times 10^{-5}$. Second, we scanned all variants annotated as functional variants with a moderate to high impact on protein function by VEP for associations with neurological and cardiovascular endpoints from FinnGen R10, including the FinnGen endpoint categories Neurological endpoints, VI Diseases of the nervous system (G6_), and IX Diseases of the circulatory system (I9_) at significance level $1 \times 10^{-4}$.

Third, we scanned all variants with PIP > 0.1 among the same FinnGen neurological and cardiovascular endpoints at significance level $1 \times 10^{-4}$.

Results can be browsed from PheWAS app https://hhautakangas.github.io/phewas_migraine_tables.html.

## Reporting summary

Further information on research design is available in the Nature Portfolio Reporting Summary linked to this article.

## Data availability

The GWAS summary statistics for UK Biobank are publicly available in GWAS Catalog under accession code GCST90671940. The access to the UK biobank data can be applied through https://www.ukbiobank.ac.uk/. The GWAS summary statistics for FinnGen R8 are publicly available through https://www.finngen.fi/en/access_results. The Finnish biobank data can be accessed through the Fingenious® services (https://site.fingenious.fi/en/) managed by FINBB. Finnish Health register data can be applied from Findata (https://findata.fi/en/data/). The GWAS summary statistics for the 23andMe data set will be made available through 23andMe to qualified researchers under an agreement with 23andMe that protects the privacy of the 23andMe participants. Please visit https://research.23andme.com/collaborate/#publication for more information and to apply to access the data. Fine-mapping results are provided in Supplementary Tables S3a, S4 and S5. PheWAS results can be browsed at https://hhautakangas.github.io/phewas_migraine_tables.html.

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

## Acknowledgements

We would like to thank the research participants and employees of 23andMe, Inc. for making this work possible. We thank all the study participants, employees, and investigators of FinnGen, UK Biobank and The Trøndelag Health Study (HUNT), for their contribution to this research. This research has been conducted using the UK Biobank Resource under Application Number 22627. This work was supported by grants no. 336825, 338507, 352795 from the Research Council of Finland to M.P., by Sigrid Jusélius foundation (M.P. and A.P.) and by the Doctoral School of University of Helsinki and the Biomedicum Helsinki Foundation, Young Investigator Grant, 2023 (H.H.). Open access funded by Helsinki University Library. The FinnGen project is funded by two grants from Business Finland (HUS 4685/31/2016 and UH 4386/31/2016) and the following industry partners: AbbVie Inc., AstraZeneca UK Ltd, Biogen MA Inc., Bristol Myers Squibb (and Celgene Corporation & Celgene International II Sàrl), Genentech Inc., Merck Sharp & Dohme LCC, Pfizer Inc., GlaxoSmithKline Intellectual Property Development Ltd., Sanofi US Services Inc., Maze Therapeutics Inc., Janssen Biotech Inc, Novartis AG, and Boehringer Ingelheim International GmbH. Following biobanks are acknowledged for delivering biobank samples to FinnGen: Auria Biobank (www.auria.fi/biopankki), THL Biobank (www.thl.fi/biobank), Helsinki Biobank (www.helsinginbiopankki.fi), Biobank Borealis of Northern Finland (https://www.ppshp.fi/Tutkimus-ja-opetus/Biopankki/Pages/Biobank-Borealis-briefly-in-English.aspx), Finnish Clinical Biobank Tampere (www.tays.fi/en-US/Research_and_development/Finnish_Clinical_Biobank_Tampere), Biobank of Eastern Finland (www.ita-suomenbiopankki.fi/en), Central Finland Biobank (www.ksshp.fi/fi-FI/Potilaalle/Biopankki), Finnish Red Cross Blood Service Biobank (www.veripalvelu.fi/verenluovutus/biopankkitoiminta), Terveystalo Biobank (www.terveystalo.com/fi/Yritystietoa/Terveystalo-Biopankki/Biopankki/) and Arctic Biobank (https://www.oulu.fi/en/university/faculties-and-units/faculty-medicine/northern-finland-birth-cohorts-and-arctic-biobank). All Finnish Biobanks are members of BBMRI.fi infrastructure (www.bbmri.fi). Finnish Biobank Cooperative -FINBB (https://finbb.fi/) is the coordinator of BBMRI-ERIC operations in Finland. The Finnish biobank data can be accessed through the Fingenious® services (https://site.fingenious.fi/en/) managed by FINBB. The Trøndelag Health Study (HUNT) is a collaboration between HUNT Research Centre (Faculty of Medicine and Health Sciences, Norwegian University of Science and Technology NTNU), Trøndelag County Council, Central Norway Regional Health Authority, and the Norwegian Institute of Public Health. The genotyping was financed by the National Institute of health (NIH), University of Michigan, The Norwegian Research council, and Central Norway Regional Health Authority and the Faculty of Medicine and Health Sciences, Norwegian University of Science and Technology (NTNU). The genotype quality control and imputation have been conducted by the K.G. Jebsen Center for Genetic Epidemiology, Department of Public Health and Nursing, Faculty of Medicine and Health Sciences, Norwegian University of Science and Technology (NTNU).

## Author contributions

H.H. and M.P. designed the study and drafted the manuscript. H.H. performed analyses, except J.K. performed Susie analysis, and M.P. performed pQTL analysis. M.P. supervised analyses and the project. H.H., J.K., A.P., and M.P. critically evaluated and revised the manuscript.

## Competing interests

A.P. is the Scientific Director of the public-private partnership project FinnGen that has 12 industry partners that provide funding for the FinnGen project. Other authors declare no competing interests.

## Additional information

## FinnGen

Aarno Palotie [ORCID] [1,2,3]

## International Headache Genetics Consortium

Heidi Hautakangas [ORCID] [1] ✉, Aarno Palotie [ORCID] [1,2,3] & Matti Pirinen [ORCID] [1,4,5] ✉

A full list of members and their affiliations appears in the Supplementary Information.

