## [Transparent Peer Review File · Nature Communications]

Fine-mapping a genome-wide meta-analysis of 98,374 migraine cases identifies 181 sets of candidate causal variants

Corresponding Author: Professor Matti Pirinen

Version 0:

Reviewer comments:

Reviewer #1

(Remarks to the Author)

This study makes a significant contribution to the understanding of the genetic basis of migraine through large-scale meta-analysis and fine-mapping techniques. I have several comments and concerns about your manuscript as follows:

1. Line 123 of the manuscript mentions the genetic correlation between different study collections, but the potential impact of these differences on the final analysis results is not fully explained. For example, the lower correlation of FinnGen with other datasets could affect the interpretation and reliability of the results.
2. The manuscript mentions several significant results, but does not describe in detail the methods used to correct for multiple comparisons. Although statistical significance is mentioned, there is no specific explanation of how the false positive rate is controlled.
3. Although the manuscript mentions the sources and merging of data, it lacks detailed descriptions of the data cleaning and quality control procedures. For example, how are missing data, outliers and potential batch effects handled?
4. The manuscript does not adequately discuss potential biases and confounding factors that could affect the results. Although LD was used in the analysis, there is no detailed explanation of how systematic differences between different data sets were accounted for.

Minor

1. It is recommended to provide a detailed explanation of the terms "UKB" and "UKB-FG" used to label Figure 3.
2. Please ensure that gene names are italicised throughout the manuscript and tables.
3. TUBG2(<https://doi.org/10.1016/j.xhgg.2023.100211>), ELAVL2(<https://doi.org/10.1093/brain/awab267>), NPP5A(<https://doi.org/10.5204/thesis.eprints.240779>) are not new risk findings. They have already been reported to be associated with migraine.

Reviewer #2

(Remarks to the Author)

Dr. Matti Pirinen and co-authors present a fine-mapping study of a large GWA meta-analysis of 98,374 migraine cases and 869,160 controls from 3 cohorts (FinnGen, UK Biobank, and 23andMe) using the LD information from the original GWAS data (i.e., the exact in-sample LD) and the FINEMAP method. They discovered new genetic loci that are associated with migraine susceptibility and prioritized potential causal genetic variants and genes. The manuscript is well-written (even if sometimes it reads more like a methodological paper), the methods and approaches used are innovative and rigorous, and the sample size is large enough to draw conclusions. Because migraine is such a common affliction, this manuscript would be an excellent resource of genetic information to interested readers of Nature Communications. Further, the results clearly add to the literature of migraine genetics. However, I have some comments and recommendations that may result in a markedly improved paper if they can be addressed.

Major points

1. I recommend conducting additional in silico analyses (e.g., TWAS, PWAS) and functional studies to better understand the role of prioritized variants and genes in migraine signaling pathways and to confirm if these fine-mapped variants and genes are truly causal.

2. In the abstract, last sentence, please highlight the biological relevance/importance that this fine-mapping study provided and make the link with migraine susceptibility. What do we learn from the current work for migraine etiology? How those potential causal genes contribute to migraine? What are their biological roles?

3. How were the 'novel' loci defined? Which distance was used to identify new loci (vs. previously reported)? Among the 35 'novel' loci reported in Table 2, two loci (SF3B4 and near MTCH2) have been previously identified in Choquet et al. *Commun Biol.* 2021 (PMID: 34294844 – please see Supplementary Data) and one gene (TUBG2) has been previously identified in Meyers et al. *HGG Adv.* 2023 (PMID: 37415806). Further, among the 7 prioritized genes with high confidence in Table 3, four genes (PHACTR1, LRP1, 1:156,409,585, and NGF) have been previously reported in both Meyers et al. and Choquet et al.

Please double check previously published GWAS of migraine to confirm the 'novel' loci. Also, please cite Meyers et al. *HGG Adv.* 2023 (PMID: 37415806) and discuss the overlap in terms of previous findings with the current work.

4. Results pages 14-15 (lines 273-275). Please explain PENC and the rationale for the cut-off of $PENC > 3$ as "low quality". What if putatively causal variants are not being considered because PENC is slightly more than 3 or their in-sample LD is not available. While $PENC > 3$ threshold was 'justified' in lines 236-240, the explanation was not clear. Are there previous articles to support the use of this threshold? Putatively causal may be excluded because of lack of available in-sample LD. It is unclear if true causal variants may have been missed to report because of lack of in-sample LD to prove the methodology. Further, there are no references for PENC (line 203) or PIP (line 179 - Figure 1 legend) or maximum difference of the variant-specific posterior inclusion probabilities (line 205-206).

5. There are some discrepancies between the PIP cut-offs for variants reported in Table 3 ($PIP > 0.5$) and 2 NTRK1 variants (rs6339 and rs6336 with PIPs of 0.48 and 0.39, respectively) that are mentioned in the Discussion (lines 295-296). The PIPs seem low to conclude those NTRK1 variants are possibly causal. Would the authors consider commenting on this apparent contradiction in the Discussion?

Minor points

1. Please specify more at front of the manuscript (in the Abstract or in the Introduction) that all individuals included in the GWA meta-analysis are all of European ancestry.

2. Introduction, page 5 (lines 100-102): "Importantly, we have the full in-sample LD available for 26 risk loci and for the remaining risk loci we have the in-sample LD for FinnGen and UKB but not for 23andMe (Table 1)". Please rephrase as you did not mention about the '26 risk loci' yet (i.e., specify which ones are those '26 risk loci' vs. 'the remaining risk loci'). In other terms, how you defined those two loci categories?

3. Page 16 (lines 319-323), please provide the names of the specific variants and genes.

4. Supplementary Figure 1, y-axis title 'Mean χ^2 ', χ appears as a box.

5. Page 11 (line 212) What is UKB-FG reference panel? Please define UKB-FG (and add a reference); how does it differ from UKB?

6. Figure 4 is hard to interpret as the legend and the color scale explanation are missing.

7. I am still confused about the interpretation of the PENC. Perhaps adding a sentence explaining that lower PENC means lower probability of variant being causal would be helpful.

8. Results, page 15 (line 282) 'Variant Effect Predictor (VEP)'. Please add a description of the purpose of conducting this look-up and a reference.

Reviewer #3

(Remarks to the Author)

I co-reviewed this manuscript with one of the reviewers who provided the listed reports. This is part of the Nature Communications initiative to facilitate training in peer review and to provide appropriate recognition for Early Career Researchers who co-review manuscripts

Reviewer #4

(Remarks to the Author)

This manuscript describes a meta-analysis of three large studies and a following fine-mapping analysis using the meta-analysis results. Given the large sample size used, the results are likely useful for studying migraine. I have the following comments.

1) As discussed in the paper, SNPs were removed because it is not presented in all three data sets, including some leading SNPs. This is worrisome because some causal variants may be removed, and the final results do not satisfy the assumption of fine mapping: all causal variants are included. A better way to deal with this is to use the high quality summary statistics within each study to impute the missing summary statistics of those missing, e.g., PMID: 24990607, 32053016. Then perform fine mapping.

2) For fine mapping of multiple population data sets, there is a recent fine mapping method XMAP that can combine multiple data sets (<https://www.nature.com/articles/s41467-023-42614-7>). This is likely to be a better option. For those loci without in-sample LD, a reference based LD matrix may still be needed. Please run XMAP and compare with the meta-analysis based fine mapping method.

3) This paper proposes to use the expected number of causal variants to classify high/low quality regions. The idea is intuitive. However, given the smaller number of known low quality regions, it is hard to know how good it is. SuSiE proposed a way to perform diagnostics when estimated LD is used (<https://journals.plos.org/plosgenetics/article?id=10.1371/journal.pgen.1010299>). Could you try to apply the diagnostics to your loci with ground truth and also loci without ground truth? Maybe the diagnostics can be more informative than the expected number of causal variants.

4) Line 141, it mentions that the LDSC intercept for FinnGen is 1.10 and it is likely due to the related samples used by REGENIE. Can the authors re-run the FinnGen using simple logistic regression using unrelated samples to verify this? This might be important to make sure the meta-analysis statistics not inflated.

5) For the validation of new loci from meta-analysis, the significance level should be 0.05/35, or FDR with threshold 0.05 can be used. Either way, it is important to adjust for number of tests.

6) Table 2. It is better to report odds ratio instead of log-odds ratio because odds ratio is easier to understand its effect size.

Version 1:

Reviewer comments:

Reviewer #1

(Remarks to the Author)

The authors have provided a comprehensive and well-structured response, and the additional analyses substantially strengthen the manuscript. Most original concerns are addressed, and the data appear robust. A few points remain:

Major

1. Details on Bonferroni correction and significance thresholds improve clarity. However, the PheWAS section uses lenient thresholds, with results proposed as “for further screening.” Please indicate in the main text or supplement how many associations remain significant under stricter global corrections (e.g., Bonferroni or FDR). Has the Bonferroni threshold accounted for correlations among analyses, and could a more flexible FDR approach improve power?

2. QC was conducted separately in the three cohorts, then harmonized using unified filters (e.g., MAF, INFO, HWE). While reasonable, could excluding multi-allelic variants remove functionally important rare variants? Please report the proportion excluded and discuss implications for fine-mapping reliability.

Minor

1. LDSC intercepts, LD scores, and PENC thresholds were added to control confounding, and the absence of rare variants was acknowledged. After LDSC adjustment, which novel loci showed the greatest loss of significance? Could some be driven by residual confounding?

2. The explanation for lower genetic correlation in FinnGen and its limited effect on effect size estimates is convincing. Still, a sensitivity analysis excluding FinnGen (UKB+23andMe only) for key loci would further test robustness.

Reviewer #2

(Remarks to the Author)

The authors have done an excellent job addressing my comments.

Reviewer #3

(Remarks to the Author)

Reviewer #4

(Remarks to the Author)

The authors have addressed my questions adequately. I have no other questions.

We thank the Reviewers for the valuable time and effort they have devoted to providing insightful comments about our work. We have now revised our paper based on the reviewers' comments, and we provide point-by-point responses below.

Note about the author list: We have added Joonas Kartau as an author for his contribution to the revision work. We have moved "HUNT All-in Headache" consortium to Acknowledgments following the journal's instructions about consortia where no member is an author of the work.

Reviewer #1 (Remarks to the Author):

This study makes a significant contribution to the understanding of the genetic basis of migraine through large-scale meta-analysis and fine-mapping techniques. I have several comments and concerns about your manuscript as follows:

1. Line 123 of the manuscript mentions the genetic correlation between different study collections, but the potential impact of these differences on the final analysis results is not fully explained. For example, the lower correlation of FinnGen with other datasets could affect the interpretation and reliability of the results.

Thank you for this observation. We have reported that the estimated genetic correlations were 1.00 (s.e. 0.04) between UKB and 23andMe, 0.84 (s.e. 0.05) between UKB and FinnGen, and 0.87 (s.e. 0.03) between 23andMe and FinnGen. A slightly reduced (< 1.0) genetic correlation between FinnGen and other data sets suggests that the effect sizes measured by FinnGen are not exactly the same as in the other studies. However, since these correlations (0.84-0.87) are still high, FinnGen is clearly measuring very similar effect sizes as the other two studies. Deviations from the perfect genetic correlation are expected already because the migraine case definition is different in FinnGen (at least one purchase of triptans) compared to the other two studies (self-reported migraine). Additionally, the Finnish genetic background (FinnGen) vs. non-Finnish European background (UKB and 23andMe) can explain some of the imperfect genetic correlation. We explain these possibilities on p.6:

The lower genetic correlation between FinnGen and the other two studies could be due to differences in the case definitions (triptan purchases in FinnGen vs. self-reporting in UKB and 23andMe). A comparable level of genetic correlation (0.81) has been reported before between primary care and self-reported migraine cases within UKB¹. Another source of possible heterogeneity in effect sizes is the difference in genetic ancestry (Finnish in FinnGen vs. Non-Finnish European in the other two).

The expected impact of imperfect genetic correlation is that the effect sizes measured by our meta-analysis are averaged effect sizes between the definitions of migraine used in our component studies. However, the forest plots (Supplementary Figure 3) and the heterogeneity test (reported on p.8) show that FinnGen and the other two studies are highly consistent across our 122 lead SNPs and thus we expect that the slightly

reduced genetic correlation between FinnGen and the other two studies have little impact on our estimates of migraine effect sizes. We explain this on p.8:

We observed statistically significant heterogeneity ($P < 0.05/122$) in effect sizes between the study collections only for two lead variants, both of which resided in the previously known migraine loci (PRDM16 and near ZCCHC14) (Supplementary Data 1, Supplementary Fig 3).

We have added to Discussion a comment about the impact of reduced genetic correlation of the results (p.26):

We observed that the genetic correlation between FinnGen and the other two data sets was slightly below 1.0 but we found little evidence for heterogeneity in the effect size estimates at the reported risk loci between the data sets.

2. The manuscript mentions several significant results, but does not describe in detail the methods used to correct for multiple comparisons. Although statistical significance is mentioned, there is no specific explanation of how the false positive rate is controlled.

We agree that reporting the definition of statistical significance is important. We have now included the following in the Methods p.30:

The false positive rate was controlled based on the significance thresholds mentioned in the text, that is, the results were considered statistically significant when the corresponding P-value was below the significance threshold. Multiple comparisons were accounted for by the Bonferroni correction.

Below we list the instances of the term “statistical significance” from the manuscript, and we list the places in the text where we explain how these significance levels were determined.

1. P.7 we require “genome-wide significance (GWS, $P < 5 \times 10^{-8}$)” to report that a lead variant of a locus is statistically significant (this applies to all our 122 loci, whether known or novel). This threshold is the norm in the GWAS field and it can be thought of as a multiplicity correction for a genome-wide set of one million independent tests that are carried in any one GWAS, i.e., $5 \times 10^{-8} = 0.05/10^6$. See, for example, a review on GWAS methodology by Uffelmann et al.² (<https://doi.org/10.1038/s43586-021-00056-9>).
2. Statistically significant heterogeneity among the 122 lead variants was defined by the Bonferroni multiple testing correction, and this multiple testing correction was explicitly stated as “ $P < 0.05/122$ ” on p.8.
3. “Significant cis-eQTL” on p.16 was defined based on the definition of significant eQTLs provided by the GTEx project and significant pQTLs provided by the UK biobank Pharma Proteomics Project. We explain these on p.36:

To assess variants' effects on gene expression, we used the list of significant eQTLs of the 49 tissues as reported by GTEx v.8 (<https://gtexportal.org/home/>). For assessing variants' effects on the plasma proteome, we used the list of significant pQTLs as reported by the UK biobank Pharma Proteomics Project in their Supplementary Table 10 and extracted the data from their website (<https://metabolomips.org/ukbbpgwas/>).

4. To collect the list of the results from the phenome-wide association studies (P.20), we used significance levels $\times 10^{-5}$ (for all top variants and all FinnGen endpoints) and 10^{-4} (for more specific set of functional variants and neurological endpoints). These thresholds were used in order to provide a reasonably sized lists about potentially interesting phenome-associations of the migraine risk variants that we could include in the supplementary material for other researchers to browse and study. The idea is that these lists can be further filtered by other researchers using more strict multiple testing corrections as relevant for any particular downstream application.

3. Although the manuscript mentions the sources and merging of data, it lacks detailed descriptions of the data cleaning and quality control procedures. For example, how are missing data, outliers and potential batch effects handled?

We agree that these are important pieces of information. As is typical to large meta-analyses, the GWAS used in our meta-analysis had been run earlier and we give the references on p.27-28 to the sources of these GWAS where the details of each GWAS analysis can be found.

- UK biobank (p. 27): *“We used the migraine GWAS data described in³ with self-reported migraine as the phenotype.”*
- 23andMe (p. 27): *“23andMe migraine GWAS was performed by a personal genomics company 23andMe, Inc. (<https://www.23andme.com/>) and detailed description of the migraine GWAS is provided elsewhere⁴.”*
- FinnGen (p. 28): *“A detailed description of FinnGen is provided in⁵. We used the 8th Data Freeze for the migraine GWAS. The migraine cases were defined as the individuals who had at least one triptan purchase and the remaining individuals without any triptan purchases were defined as controls from the social insurance institution of Finland (KELA) registry including medication reimbursement and drug purchases (https://r8.ristey.s.finnngen.fi/phenocode/MIGRAINE_TRIPTAN).”*

Outliers and potential batch effects were handled first by each of the above mentioned three source GWAS depending on what they considered suitable in their setting.

Additionally, we further applied common QC criteria, including a missingness filter, for all the GWAS results from the 3 sources as explained on p. 31:

We excluded multi-allelic variants, and variants with minor allele frequency (MAF) < 0.01, IMPUTE2 info or MACH r2 < 0.6, and when available, missingness > 0.05 and Hardy-Weinberg equilibrium (HWE) $P < 1 \times 10^{-6}$ from each study.

4. The manuscript does not adequately discuss potential biases and confounding factors that could affect the results. Although LD was used in the analysis, there is no detailed explanation of how systematic differences between different data sets were accounted for.

We agree that it is important to clearly acknowledge potential biases and confounding factors of GWAS meta-analyses. First, there are potential biases related to the GWAS approach due to genetic relatedness and population stratification. We have assessed these by using LD score regression intercept parameter. We acknowledge an elevated value for the LDSC intercept on p.6:

However, as the intercept from LDSC was elevated to 1.09 (s.e. 0.01) from its null value of 1.0, some inflation could also be due to confounding factors such as cryptic relatedness, population stratification or other model misspecification, which we will take into account when reporting new loci.

Additionally, we have now revised the results in Table 2 so that the effect of a correction by the LDSC intercept can be seen for the novel loci. We mention on p.7:

After genomic control by the LDSC intercept, 22 novel loci remained GWS (Table 2).

and explain how this was done in Methods on p.32.

The intercept of the LD score regression analysis can be used as a genomic control parameter to provide a conservative P-value by dividing the observed chi-square statistic of the GWAS association by the intercept before computing the P-value from the chi-square distribution⁶. We used this approach to indicate which variants in Table 2 had $P > 5 \times 10^{-8}$ after the genomic control.

When it comes to the LD, the systematic differences between the data sets in LD structure were accounted for by weighting the LD of each study by its effective sample size. This way the combined LD matches with the inverse-variance weighted GWAS meta-analysis summary statistics and we have accurate in-sample LD for the meta-analysis. This in-sample LD option was available in 26 of our loci, where we expect to have very high-quality results. For the remaining loci, we used the UKB's LD as a proxy for 23andMe's LD. We studied in detail how this approximation worked, and our conclusion is that it works well as long as the PENC statistic was not above 3.0. We report this on p.14:

To define suitable thresholds for PENC, we observed that all low-quality regions (defined as $\max\Delta > 0.1$) had $PENC > 3$ with the UKB-FG panel (Fig 3b). Thus, we expect that this threshold has a high sensitivity to filter out the low-quality results in our analyses. The exact value of a reliable PENC threshold depends on the accuracy of the LD panel. For example, in our case, the less accurate UKB LD panel would require a more stringent threshold of $PENC > 2$ (Fig 3b). Previously, a PENC value of 3.5 has been used in a schizophrenia GWAS fine-mapping with a high-quality LD panel⁷.

Finally, there are potential biases due to limitations in the coverage of genetic variation included in our fine-mapping. We have discussed these on p.24-25

Our study has some limitations. First, since reliable fine-mapping requires that we exclude variants that are not present in all three component studies of our meta-analysis, it is possible that we exclude also some of the true causal variants. This is a potential problem especially when some of the top variants of the fine-map region have been filtered out from fine-mapping.

Additionally, since very rare variants were not included in our analysis, we miss the true causal variants that are rare. Since our variant set is not comprehensive, we must keep in mind that also variants that have a very high probability of being causal in our analysis may still have such variants in high LD that were not included in our analysis.

Minor

1. It is recommended to provide a detailed explanation of the terms "UKB" and "UKB-FG" used to label Figure 3.

We have updated the legend of Figure 3 to explain these abbreviations:

a) Scatter plot comparing the maximum PIP differences ($\max\Delta$) between the in-sample and reference LD for 26 fine-map regions, where reference LD is based on UK biobank + FinnGen (UKB-FG) or only UK biobank (UKB).

2. Please ensure that gene names are italicised throughout the manuscript and tables.

We have now italicized gene names throughout the text and Tables 2 & 3. Please note that in some places we mean the protein rather than the gene and there we have not italicized the protein name.

3. TUBG2(<https://doi.org/10.1016/j.xhgg.2023.100211>), ELAVL2(<https://doi.org/10.1093/brain/awab267>), NPP5A(<https://doi.org/10.5204/thesis.eprints.240779>) are not new risk findings. They have already been reported to be associated with migraine.

Thank you for providing links to these previous findings that we have now incorporated in the manuscript text.

The *TUBG2* gene was identified in a TWAS (Meyers et al 2023)⁸ that studied whether genetically predicted gene expression associates with a migraine GWAS signal. The *ELAVL2* locus was reported (Bahrami et al. 2022)⁹ using a joint analysis of depression and migraine with conditional joint false discovery approach. We now acknowledge these previous findings in the manuscript P.7

Of the new loci, TUBG2 has been implicated in a transcriptome-wide association study on migraine⁸ and ELAVL2 in a joint analysis of depression and migraine⁹.

As we have no variants in these loci that have been reported in a previous migraine GWAS using the standard GWAS criterion of $P < 5 \times 10^{-8}$, we still keep these loci in our Table 2. For this check, we used the same locus definition as was used in the previous large migraine meta-analysis by Hautakangas et al. 2022 and defined a locus as new if the high LD region, i.e., the region formed by the lead variant and all variants in high LD with the lead variant, +-250 kb did not overlap any previously reported risk variant from a previous migraine GWAS. We have made our locus novelty check based on 11 previous migraine GWAS (our references 4-14).

We agree that the *INNP5A* locus is a known locus since it has been already reported, for example, by Hautakangas et al. 2022. We have listed it as a known locus in our results and not as a new locus.

Reviewer #2 (Remarks to the Author):

Dr. Matti Pirinen and co-authors present a fine-mapping study of a large GWA meta-analysis of 98,374 migraine cases and 869,160 controls from 3 cohorts (FinnGen, UK Biobank, and 23andMe) using the LD information from the original GWAS data (i.e., the exact in-sample LD) and the FINEMAP method. They discovered new genetic loci that are associated with migraine susceptibility and prioritized potential causal genetic variants and genes. The manuscript is well-written (even if sometimes it reads more like a methodological paper), the methods and approaches used are innovative and rigorous, and the sample size is large enough to draw conclusions. Because migraine is such a common affliction, this manuscript would be an excellent resource of genetic information to interested readers of Nature Communications. Further, the results clearly add to the literature of migraine genetics. However, I have some comments and recommendations that may result in a markedly improved paper if they can be addressed.

Major points

1. I recommend conducting additional in silico analyses (e.g., TWAS, PWAS) and functional studies to better understand the role of prioritized variants and genes in migraine signaling pathways and to confirm if these fine-mapped variants and genes are truly causal.

Thank you for these recommendations. We are now including the associations for the variants prioritized by fine-mapping (PIP > 0.1) with gene expression across 49 tissues based on GTEx data (Supplementary Data 4) and plasma proteome based on UKB Pharma Proteomics Project (Supplementary Data 7). We report the main results also in the main text:

p.16

rs6330 is a significant cis-eQTL for NGF-AS1 expressed in atrial appendage of heart and rs28929474 for IFI27L2 expressed in tibial artery and in left ventricle of heart in GTEx v.08 data.

p.17

The missense variant rs28929474 is highly pleiotropic showing associations to 132 proteins (Supplementary Data 7)

p.18

the candidate variant in LRP1 is associated with two proteins (Supplementary Data 7) and several vascular diseases, such as sporadic thoracic aortic dissection, fibromuscular dysplasia and spontaneous coronary artery dissection^{10,11,12}.

We agree that also functional studies will be needed to determine the exact biological mechanisms related to associations and we hope that our results will stimulate such work in near future. However, we feel that planning and executing such studies fall outside the scope of our current manuscript that already introduces and empirically evaluates a practical way to do fine-mapping in GWAS meta-analyses and provides detailed fine-mapping results for migraine.

2. In the abstract, last sentence, please highlight the biological relevance/importance that this fine-mapping study provided and make the link with migraine susceptibility. What do we learn from the current work for migraine etiology? How those potential causal genes contribute to migraine? What are their biological roles?

Thank you for the suggestions to emphasize the biological understanding of the findings. The biological interpretation of the variants seems simplest for the missense variants. In the Abstract, we report 2 missense variants with high PIPs in our fine-mapping pointing to the genes *NGF* and *INPP5A*. We know, for example, that NGF and its receptors have a central role in the pain perception, and elevated NGF levels have been observed in chronic pain conditions. However, since our Abstract is at its maximum length (200 words), we would prefer to explain what is known about these genes later in the main text, p.22-23 (lines 433-472):

Our high-quality results highlight two missense variants with high PIPs: rs6330 (PIP=0.59) in NGF and rs1133400 (PIP=0.93) in INPP5A.

The variant rs6330 is only in weak LD ($r^2 = 0.04$) with the lead variant (rs12134493) of its locus and was identified as a secondary signal in our fine-mapping. A recent study¹³ has also reported that the migraine association of rs6330 remained statistically significant in a conditional analysis after adjusting for the stronger signal (rs2078371) within the same risk locus. NGF has been reported to be highly expressed in hippocampus and cortex^{14,15} although according to the GTEx v8 data, NGF does not show statistically significant expression in any brain tissue but shows high expression in multiple other tissues, including, for example, ovary, tibial nerve, arteries, visceral adipose, and heart. NGF levels have been reported to be elevated in cerebrospinal fluid in chronic migraine patients compared to controls¹⁶, and decreased in blood serum of episodic migraine patients compared to controls and chronic migraine patients¹⁷. The potential causality of the pathway related to NGF was further supported by two additional missense variants, rs6339 (PIP=0.48) and rs6336 (PIP=0.39), located in NTRK1 which encodes one of the two receptors for NGF. NGF and its receptors have a central role in the pain perception, and elevated NGF levels have been observed also in many other chronic pain conditions, such as osteoarthritis and low back pain^{18,19,20}. Multiple antibodies of NGF or small molecular inhibitors of the NGF receptors have been developed and tested in clinical studies to treat chronic pain conditions, including low back pain and osteoarthritis^{21,22,23,24,25}. Even though some candidate drugs have shown potential benefit relating to pain relief, an increased risk of progressive osteoarthritis has been observed in a small group of the treated patients²⁵, and therefore none of the drugs have yet received FDA approval. Currently, other type of drug classes (p75 neurotrophin receptor fusion protein, LEVI-04 (ClinicalTrials.gov ID: NCT05618782) and anti-NGF PEGylated Fab' antibody²⁶), are being developed and in pre-clinical or clinical testing. In adults, after pain stimuli, NGF activates overexpression of other neuronal molecules, including calcitonin gene-related peptide (CGRP) and substance P²⁵. CGRP is involved in migraine pain, and several effective monoclonal antibodies targeting either CGRP or its receptors have been developed to treat migraine^{27,28,29}. Gene INPP5A is highly expressed in Purkinje cells of cerebellum³⁰ and involved in multiple cellular signaling processes including neurotransmission, hormone secretion, cell proliferation and muscle contraction through its role in the pathway regulating intracellular calcium levels. The missense variant rs1133400 is in modest LD ($r^2 = 0.36$) with the lead variant of the locus (rs200314499) that was filtered out from the fine-mapping in the quality control phase. For this locus, FINEMAP suggested two causal variants (PENC = 1.65). PheWAS showed no other associations with this missense variant.

We will also go through in detail other prioritized genes in the main text, such as SERPINA1 (p.17-18 l.327-328):

SERPINA1 encodes an alpha-1 antitrypsin, a serine protease inhibitor protein, that belongs to the serpin superfamily. Its primary target is elastase, and other targets are plasmin and thrombin. Several mutations, including our high-PIP variant rs28929474C>T, in SERPINA1 can cause an autosomal co-dominant genetic disorder alpha-1 antitrypsin (AAT) deficiency, which can lead to lung or liver disease due to reduced alpha-1 antitrypsin levels³¹. The missense

variant rs28929474 is highly pleiotropic showing associations to 132 proteins (Supplementary Data 7) and multiple disease categories in PheWAS of FinnGen R10 data including, for example, diseases of the respiratory system, diseases of the circulatory system, diseases of digestive system, pregnancy related diseases, diseases of the nervous system, and diseases of musculoskeletal system and connective tissue (Supplementary Data 6,8,9).

LRP1 (p.18-19 l.356-362):

the candidate variant in LRP1 is associated with two proteins (Supplementary Data 7) and several vascular diseases, such as sporadic thoracic aortic dissection, fibromuscular dysplasia and spontaneous coronary artery dissection^{10,11,12}. The LDL receptor-related protein 1 (LRP1) is a cell surface receptor and has an important role in vascular and blood brain barrier integrity^{32,33,34}. It is expressed in almost every tissue, and most studied in liver and brain. LRP1 is also involved in vascular calcium signaling by regulating smooth muscle cell contractility³³. A recent study suggested that LRP1 expression is regulated by allele-specific mechanism of intronic rs11172113 located in an enhancer region through two transcription factors (MECP2 and SNAIL)³⁵.

and PHACTR1 (p.23-24 l.474-494):

Another important finding is in the PHACTR1 locus, which is one of the strongest known migraine risk loci. There, our fine-mapping suggested one causal variant (PENC = 1.29), with the lead variant rs9349379 being a clear candidate for being causal with PIP of 1.00, consistent with previous results³⁶. In our FinnGen PheWAS, we detected also strong associations between the variant and, for example, major coronary disease events ($P = 8.22 \times 10^{-52}$), ischemic heart disease ($P = 1.18 \times 10^{-38}$) and angina pectoris ($P = 7.71 \times 10^{-26}$), all to the opposite directions compared to migraine risk. Because of these well-known associations with multiple vascular diseases, this locus has been previously studied in detail but with contradicting results. Gupta et al. (2017)³⁷ reported that rs9349379 regulates upstream gene EDN1, whereas Wang et al. (2018)³⁸ reported that they failed to replicate this endothelial rs9349379-EDN1 eQTL, but instead showed that rs9349379 regulates the closest gene PHACTR1, confirming previous vascular rs9349379-PHACTR1 eQTLs. Further, Rubin et al (2022)³⁹ observed that a loss of PHACTR1 gene does not seem to have any effect on the endothelial or smooth muscle cells of the transgenic mice, and suggested that PHACTR1 has no contribution to pathological vascular phenotype in mice through cells involved in vascular physiology. Our fine-mapping has provided strong evidence that the lead variant rs9349379 is causal for migraine, but given that the variant is intronic, our fine-mapping results alone do not provide direct evidence through which gene or mechanism this association affects the disease risk.

3. How were the 'novel' loci defined? Which distance was used to identify new loci (vs.

previously reported)? Among the 35 ‘novel’ loci reported in Table 2, two loci (SF3B4 and near MTCH2) have been previously identified in Choquet et al. Commun Biol. 2021 (PMID: 34294844 – please see Supplementary Data) and one gene (TUBG2) has been previously identified in Meyers et al. HGG Adv. 2023 (PMID: 37415806). Further, among the 7 prioritized genes with high confidence in Table 3, four genes (PHACTR1, LRP1, 1:156,409,585, and NGF) have been previously reported in both Meyers et al. and Choquet et al.

Please double check previously published GWAS of migraine to confirm the ‘novel’ loci. Also, please cite Meyers et al. HGG Adv. 2023 (PMID: 37415806) and discuss the overlap in terms of previous findings with the current work.

Thank you for providing links to these previous findings.

We used the same locus definition as was used in Hautakangas et al. 2022 and defined a locus as new if the high LD region (i.e. region formed by the lead variant and all variants in high LD with the lead variant) +/-250 kb did not overlap any previously reported migraine risk variant. We have made our locus novelty check based on 11 previous migraine GWAS (references 4-14 in the main manuscript).

We have now clarified this in the text p.7:

We followed the locus definition of Hautakangas et al. (2022) and defined the LD-independent genome-wide significant (GWS; $P < 5 \times 10^{-8}$) risk loci from the meta-analysis iteratively by choosing the variant with the smallest P-value as an index variant and excluding all other GWS variants with $LD r^2 > 0.1$ to that index variant from further considerations until no GWS variants remained. Next, we formed a high LD region around each index variant extending to the level of $r^2 > 0.6$, and merged regions that were closer than 250 kb. Lastly, all other GWS variants were included in their closest region, and the region boundaries were updated, and once again regions closer than 250 kb were merged (see further details in Methods). Based on this locus definition, we identified 122 LD-independent risk loci. Of these, 35 were new (Table 2), defined by no previously reported migraine risk variant^{40,41,42,43,4,44,45,46,36,3,13} residing within 250kb from the locus boundaries (Fig 1, Supplementary Data 1, Supplementary Figs 2-4).

The high LD region for the SF3B4 locus was 1:149,897,217-149,897,217. The closest reported variant in Choquet et al. 2021 was 1:150,510,660 (rs6693567) which is 613,443 base pairs from the closest endpoint of our SF3B4 LD region exceeding the 250 kb window we are using to define the loci. Please note that in Supplementary Data 1 we have reported rs6693567 as the lead variant of a known locus “near ADAMTSL4”, which is a different locus from SF3B4.

The high LD region for the locus near MTCH2 was 11:47,375,193-47,932,666. The closest reported variant in Choquet et al. 2021 was 11:46,929,954 which is 445,239 base pairs from the closest endpoint of the high LD region residing outside the 250kb window that defines a locus.

The *TUBG2* gene was identified in a TWAS (Myers et al 2023) that studied whether genetically predicted gene expression associates with migraine GWAS signal. We have now added the following information on p.7;

Of the new loci, TUBG2 has been implicated in a transcriptome-wide association study on migraine⁸ and ELAVL2 in a joint analysis of depression and migraine⁹.

As we have no variant in this locus that has been reported in a previous migraine GWAS with the standard GWAS criterion of $P < 5 \times 10^{-8}$, we still keep this locus in Table 2 that is about new loci based on GWAS evidence.

In our Table 3, the 4 associations mentioned by the Reviewer (*PHACTR1*, *LRP1*, 1:156,409,585, and *NGF*) are indeed all in previously known loci, namely in our loci named as *PHACTR1*, *LRP1*, near *ADAMSTL4* and near *TSPAN2*, respectively. Thank you for pointing out that for *PHACTR1* and *LRP1*, Choquet et al. (2021) have reported these same variants as highly probably causal (PIP > 0.95). We have now added the following text on p.18.

Our results added information on two of the strongest known migraine risk loci by estimating PIPs of 1.00 for the intronic variants rs9349379 in PHACTR1 and rs11172113 in LRP1, both of which have been previously prioritized³⁶.

And on p.23

There our fine-mapping suggested one causal variant (PENC = 1.29), with the lead variant rs9349379 being a clear candidate for being causal with PIP of 1.00, consistent with previous results³⁶.

4. Results pages 14-15 (lines 273-275). Please explain PENC and the rationale for the cut-off of $PENC > 3$ as “low quality”. What if putatively causal variants are not being considered because PENC is slightly more than 3 or their in-sample LD is not available. While $PENC > 3$ threshold was ‘justified’ in lines 236-240, the explanation was not clear. Are there previous articles to support the use of this threshold? Putatively causal may be excluded because of lack of available in-sample LD. It is unclear if true causal variants may have been missed to report because of lack of in-sample LD to prove the methodology. Further, there are no references for PENC (line 203) or PIP (line 179 - Figure 1 legend) or maximum difference of the variant-specific posterior inclusion probabilities (line 205-206).

Thank you for the comment. We have now explained PENC also in less technical language when it is introduced the first time on p.11. PENC is a value reported by the FINEMAP software, which we now say explicitly. Additionally, we also mention a previous study on schizophrenia where a high PENC value has been used to filter fine-mapping results.

FINEMAP software outputs a posterior expectation of the number of causal variants (PENC), that is an estimate of number of independent causal variants in the region fine-mapped. Previously, PENC has been used to filter FINEMAP results in the schizophrenia fine-mapping study⁷, and we chose it as one of our candidate statistics.

We have rewritten the part that explains the logic of PENC and how the threshold was determined. We have also given reference to Trubetskoy et al. (2022) that used PENC value p.13-14:

We then investigated how well the five different statistics could separate the regions with low-quality fine-mapping results from those with high-quality results for the UKB-FG. We observed that the PENC measure was able to retain the largest number of high-quality regions at the threshold where the low-quality regions were filtered out (Supplementary Fig 5,6). The reason why a high value of PENC can indicate possible problems with fine-mapping is because a mismatch between the reference LD and GWAS results often leads to several spurious signals in fine-mapping.

To define suitable thresholds for PENC, we observed that all low-quality regions ($\max\Delta > 0.1$) had $PENC > 3$ with the UKB-FG panel (Fig. 3b). Thus, we expect that this threshold has a high sensitivity to filter out the low-quality results in our analyses. The exact value of a reliable PENC threshold depends on the accuracy of the LD panel. For example, in our case, the less accurate UKB LD panel would require a more stringent threshold of $PENC > 2$ (Fig 3b). Previously, a value of 3.5 has been used in fine-mapping a schizophrenia GWAS with a high-quality LD panel⁷.

What if putatively causal variants are not being considered because PENC is slightly more than 3 or their in-sample LD is not available?

PENC is a statistic computed during fine-mapping for the whole fine-mapped region rather than separately for individual variants. Thus, we do not exclude any individual variants based on the PENC value but rather we label the fine-mapping of a certain region to be unreliable if PENC of that FINEMAP run is > 3 with the UKB-FG LD panel.

Further, there are no references for PENC (line 203) or PIP (line 179 - Figure 1 legend) or maximum difference of the variant-specific posterior inclusion probabilities (line 205-206).

As explained above, we now mention both FINEMAP and Trubetskoy et al. (2022) when we introduce PENC the first time (p.11).

Posterior inclusion probability (PIP) is a common term in the fine-mapping literature to denote the probability that a variant is included in the set of causal variants of the locus, and it is also part of the output of common fine-mapping methods (for example, FINEMAP and SuSiE). We have explicitly written “posterior inclusion probability (PIP)”

when PIP was first introduced in the Abstract on p.2 as well as in the Figure and Table legends.

Maximum difference of the variant-specific posterior inclusion probabilities ($\max\Delta$) is a value we have defined to quantify how close to fine-mapping results on the same region are to each other. We have defined it on p.11-12.

We used the maximum difference of the variant-specific posterior inclusion probabilities ($\max\Delta$) between the reference LD and the in-sample LD to assess the quality of the reference LD results in the 26 regions where the in-sample LD was available. A small $\max\Delta$ value (close to 0) indicates high quality (the reference LD produces similar results to the in-sample LD), and a large value (close to 1) indicates low quality (the reference LD produces different results from the in-sample LD).

We do not know of previous uses of this quantity in literature.

5. There are some discrepancies between the PIP cut-offs for variants reported in Table 3 (PIP > 0.5) and 2 NTRK1 variants (rs6339 and rs6336 with PIPs of 0.48 and 0.39, respectively) that are mentioned in the Discussion (lines 295-296). The PIPs seem low to conclude those NTRK1 variants are possibly causal. Would the authors consider commenting on this apparent contradiction in the Discussion?

Thank you for this observation that requires some further clarification. Once the missense variant rs6330 with PIP 0.59 has indicated *NGF* as the most probable causal gene in its locus, it becomes more probable to us that other genes closely biologically related to *NGF* could also affect migraine susceptibility through the same pathway. Therefore, we find it very interesting that we have two additional missense variants in a related receptor gene *NTRK1* (located in a locus different from *NGF*) with considerably high PIPs > 0.39. We think that together the data on these 3 variants give more evidence also for the causality of *NGF* than the result on rs6330 alone. We have now modified the Discussion to better reflect this idea on p.22:

The potential causality of the pathway related to NGF was further supported by two additional missense variants, rs6339 (PIP=0.48) and rs6336 (PIP=0.39), located in NTRK1 which encodes one of the two receptors for NGF. NGF and its receptors have a central role in the pain perception, and elevated NGF levels have been observed also in many other chronic pain conditions, such as osteoarthritis and low back pain^{18,19,20}.

Minor points

1. Please specify more at front of the manuscript (in the Abstract or in the Introduction) that all individuals included in the GWA meta-analysis are all of European ancestry.

We have added this information on p.5 in the following sentence

we conducted a migraine meta-analysis with 98,374 migraine cases and 869,160 controls of European genetic ancestry by combining data from three sources: 23andMe, Inc., FinnGen, and UK Biobank (UKB).

2. Introduction, page 5 (lines 100-102): “Importantly, we have the full in-sample LD available for 26 risk loci and for the remaining risk loci we have the in-sample LD for FinnGen and UKB but not for 23andMe (Table 1)”. Please rephrase as you did not mention about the ‘26 risk loci’ yet (i.e., specify which ones are those ‘26 risk loci’ vs. ‘the remaining risk loci’). In other terms, how you defined those two loci categories?

We were able to get LD from 23andMe for some known migraine loci before the new loci from our current analysis or from the earlier Hautakangas et al. 2022 results were available but we have not been able to have additional LD for any of the more recent loci from 23andMe. We now state explicitly on p.5 that the 26 loci were previously known loci.

Importantly, we have the full in-sample LD available for 26 of the previously known migraine risk loci whereas for the remaining risk loci we have access to the in-sample LD for FinnGen and UKB but not for 23andMe (Table 1).

3. Page 16 (lines 319-323), please provide the names of the specific variants and genes.

We have now provided these on p.18

Five additional high-impact variants on protein function were among the credible sets (stop-gained rs5758511 in CENPM, start-lost rs3825080 in ARHGAP9 and rs798488 in GNA12 and splice-acceptor rs41298712 in ENDOV and rs9906358 in TSPAN10), but only with modest PIPs below 0.01 (Supplementary Data 5),

4. Supplementary Figure 1, y-axis title ‘Mean χ^2 ’, χ appears as a box.

Thank you for pointing this out, we have now fixed the titles.

5. Page 11 (line 212) What is UKB-FG reference panel? Please define UKB-FG (and add a reference); how does it differ from UKB?

UKB-FG means reference panel that combines UK biobank and FinnGen data. We have now explicitly said this at the first use of the term on p.11

Figure 2 demonstrates this problem at the locus around TSPAN2 where fine-mapping using the in-sample LD disagrees strongly with the UKB reference LD but agrees well with a more accurate UK biobank + FinnGen (UKB-FG) reference LD.

6. Figure 4 is hard to interpret as the legend and the color scale explanation are missing.

Thank you for pointing out the missing legend, we have now added following legend to Figure 4:

Figure 4. Summary of the FINEMAP results across the 102 migraine risk regions. a) Quality of the fine-mapping with in-sample LD and reference LD from UK biobank + FinnGen. b) Number of distinct signals for the fine-mapped regions. One signal at 43 loci and 2-5 signals at 59 loci. c) Number of variants in 95% credible sets: eight signals were fine-mapped to a single variant. d) Distribution of the posterior inclusion probabilities (PIP) of the variants in credible sets. Ten variants have PIP > 90%.

7. I am still confused about the interpretation of the PENC. Perhaps adding a sentence explaining that lower PENC means lower probability of variant being causal would be helpful.

We have now explicitly stated the logic of using PENC as a candidate measure of reliability of the fine-mapping: Higher PENC suggested potential problems (p.13)

We observed that the PENC measure was able to retain the largest number of high-quality regions at the threshold where the low-quality regions were filtered out (Supplementary Fig 5,6). The reason why a high value of PENC can indicate possible problems with fine-mapping is because a mismatch between the reference LD and GWAS results often leads to several spurious signals in fine-mapping.

8. Results, page 15 (line 282) 'Variant Effect Predictor (VEP)'. Please add a description of the purpose of conducting this look-up and a reference.

VEP provides a prediction of the functional consequence of a given SNP. We have now modified the sentence to reflect this on p.16:

We conducted a look-up from Variant Effect Predictor (VEP) database for all credible sets to search for variants that are predicted to have functional consequences.

We provide the link to VEP in the Methods on p. 35:

We searched for coding variants among the CS from the Ensembl Variant Effect Predictor (VEP) (http://grch37.ensembl.org/Homo_sapiens/Tools/VEP) database by using a default of 5 kb window around the index variant.

Reviewer #3 (Remarks to the Author):

I co-reviewed this manuscript with one of the reviewers who provided the listed reports. This is part of the Nature Communications initiative to facilitate training in peer review and to provide appropriate recognition for Early Career Researchers who co-review

manuscripts

Reviewer #4 (Remarks to the Author):

This manuscript describes a meta-analysis of three large studies and a following fine-mapping analysis using the meta-analysis results. Given the large sample size used, the results are likely useful for studying migraine. I have the following comments.

1) As discussed in the paper, SNPs were removed because it is not presented in all three data sets, including some leading SNPs. This is worrisome because some causal variants may be removed, and the final results do not satisfy the assumption of fine mapping: all causal variants are included. A better way to deal with this is to use the high quality summary statistics within each study to impute the missing summary statistics of those missing, e.g., PMID: 24990607, 32053016. Then perform fine mapping.

Thank you for pointing out this issue. We agree that our fine-mapping analysis may miss some true causal variants because we have not been able to analyze those SNPs that are not present in all 3 data sets. We explicitly say this in Discussion (see the excerpt below). Unfortunately, when one of our data sets misses a particular SNP, then we also miss the LD information between that SNP and the other SNPs in that data set. For example, we do not have reliable LD information from the Finnish population for the variants that are not already imputed in the FinnGen data set as FinnGen is using the current best practices based on the most up-to-date Finnish sequencing data. Summary statistics imputation based on inaccurate reference LD could lead to misleading fine-mapping results because we could not be sure how the inaccuracies in summary statistics of the imputed SNPs affected the fine-mapping results of the whole region. We have now explained this in Discussion (p.25):

Since our variant set is not comprehensive, we must keep in mind that also variants that have a very high probability of being causal in our analysis may still have such variants in high LD that were not included in our analysis. A valid calibration of the PIPs would require that all potential causal variants were included in the analysis.

In practice, for common variants, this would require comprehensively imputed data sets with no missing variants in any of the meta-analyzed studies, and, for rare variants, availability of high coverage sequencing data. Currently, we do not yet have such resources available in typical GWAS meta-analyses of common diseases such as migraine. Summary statistics imputation methods^{47,48} could, in principle, complete the GWAS results in the individual data sets before the meta-analysis, but as the imputation would use reference LD rather than in-sample LD, the reliability of the subsequent fine-mapping results would still remain unclear.

2) For fine mapping of multiple population data sets, there is a recent fine mapping method XMAP that can combine multiple data sets

(<https://www.nature.com/articles/s41467-023-42614-7>). This is likely to be a better option. For those loci without in-sample LD, a reference based LD matrix may still be needed. Please run XMAP and compare with the meta-analysis based fine mapping method.

Thank you for pointing us to a potential application of XMAP on these data.

We set up a pipeline to run XMAP on the same summary statistics and computing environment as what we have used for running FINEMAP. XMAP ran successfully on 2 out of the 26 regions where we had in-sample LD data while the remaining 24 regions could not be run due to memory issues.

The comparison between XMAP and FINEMAP for the two regions available is shown below. For the *LRP1* locus, the results are essentially the same between the methods and both methods label rs11172113 as the causal variant with a PIP very close to 1. For the *RP11-410N8.4* locus (rs6058750) the results are highly correlated but there are some differences in the exact PIP values between the methods. We interpret these results as XMAP mainly agreeing with our FINEMAP approach in ranking the SNPs based on PIPs but there clearly are some differences at the second locus. An interesting topic for future methodological work would be to compare the approaches through comprehensive simulations and real data settings where both methods are applicable.

Fig. R1. Comparison of posterior inclusion probabilities (PIPs) of FINEMAP (x-axis) and XMAP (y-axis) in the two fine-mapped loci where it was possible to run XMAP.

We have added a note about extensions of fine-mapping to cross-populations setting and cite XMAP on p.26.

In future, methods that explicitly model both the heterogeneous effect sizes and LD patterns across the data sets may improve the locus discovery in GWAS meta-analyses⁴⁹.

3) This paper proposes to use the expected number of causal variants to classify high/low quality regions. The idea is intuitive. However, given the smaller number of known low quality regions, it is hard to know how good it is. SuSiE proposed a way to perform diagnostics when estimated LD is used (<https://journals.plos.org/plosgenetics/article?id=10.1371/journal.pgen.1010299>). Could you try to apply the diagnostics to your loci with ground truth and also loci without ground truth? Maybe the diagnostics can be more informative than the expected number of causal variants.

Thank you for proposing this complementary metric for evaluating the quality of the summary statistic. The SuSiE paper (Zou et al. 2022)⁵⁰ proposed to estimate a regularization parameter 's' that maximizes the consistency between the regularized LD-matrix $R(s)$ and the observed z-scores (formula 24 in Zou et al. 2022; Note: 'lambda' is called 's' in the SuSiE's R implementation). The idea is that if the input LD-matrix (corresponds to $s = 0$) matches well with the input z-scores, then the estimated 's' will remain close to 0, whereas if there were a more prominent mismatch between the LD-matrix R and the z-scores, then the estimated 's' becomes larger.

As suggested by the reviewer, we have estimated the 's' values for the 26 regions where we had the in-sample LD matrices to evaluate how well the reference panels worked. We used function `estimate_s_rss()` from the R package `susieR`.

First, we noticed that, as expected, the estimated 's' values were very small for in-sample LD-matrices but increased considerably when reference LD panels were used, and the less accurate reference panel (UKB) gave larger 's' values than the more accurate reference panel (UKB-FG). (See Figure R2 below.)

Fig. R2. Distribution of the regularization parameter 's' computed with `estimate_s_rss()` function from the `susieR` package for the 26 regions where the in-sample LD was available. The distribution is shown for three sources of the reference LD: in-sample LD, UK biobank LD (UKB) and UKB + FinnGen LD (UKBFG).

Next, we plotted the estimated 's' values for the 26 regions against the corresponding $\max\Delta$ values from our analyses to evaluate whether the magnitude of estimated 's' is a useful indicator of the the quality of the reference LD based fine-mapping results as measured by $\max\Delta$. (Reminder: Large $\max\Delta$ means bad quality – small $\max\Delta$ means good quality.)

Figure R3. Quality of the reference LD fine-mapping as a function of regularization parameter 's'. $\max\Delta$ on y-axis is the maximum absolute difference of the variant-wise posterior inclusion probabilities (PIPs) between the accurate fine-mapping using the in-sample LD and the approximate fine-mapping using the LD reference panel. Results with $\max\Delta > 0.1$ are considered low quality. The regularization parameter 's' on x-axis is computed with the `estimate_s_rss()` function from the `susieR` package. Results are shown for the 26 regions where the in-sample LD was available. On left, results for the UK biobank LD reference (UKB) and on right for UKB + FinnGen LD reference (UKB-FG). If we choose a threshold value for 's' that filters out both low quality regions ($\max\Delta > 0.1$) for UKB-FG LD setting, we will also filter out 13 high quality regions ($\max\Delta < 0.1$).

If we used 's' to filter out both regions with bad quality ($\max\Delta > 0.1$) present when the UKB-FG LD reference was used, then we would lose 13 good quality regions. When we did the corresponding filtering with the PENC statistic, we only lost 2 good quality regions (See Fig. 3b in the main text.) We conclude that the PENC statistic seems to better distinguish the bad quality regions (large $\max\Delta$) from the good quality regions (small $\max\Delta$). We speculate that this may be because the PENC statistic is defined from the actual fine-mapping analysis whereas the 's' value does not directly measure the consequences that potential inconsistencies in the input data have on the actual fine-mapping results. Indeed, we could have some overall inconsistencies between the LD matrix and the GWAS z-scores leading to large 's' values but if those inconsistencies occurred at such variants that they did not affect the overall fine-mapping results, then fine-mapping may still be reliable and $\max\Delta$ remains small. In contrast, PENC would not get elevated strongly if the possible inconsistencies in the data did not affect the fine-mapping results compared to the in-sample results.

We have now added Figure R3 as Supplementary Figure 6, and we have added the following text to the manuscript, p.11

The fifth statistic was a general measure of consistency between the LD matrix and GWAS results implemented in the susieR software ⁵⁰.

We refer to Supplementary Figure 6 on p.13

We observed that the PENC measure was able to retain the largest number of high-quality regions at the threshold where the low-quality regions were filtered out (Supplementary Fig 5,6).

And give more details in Methods on p.35:

The fifth statistic was the regularization parameter of Zou et al.⁵⁰ that reflects the overall consistency between the LD-matrix and the GWAS z-scores and was computed using the function estimate_s_rss() from the R package susieR.

4) Line 141, it mentions that the LDSC intercept for FinnGen is 1.10 and it is likely due to the related samples used by REGENIE. Can the authors re-run the FinnGen using simple logistic regression using unrelated samples to verify this? This might be important to make sure the meta-analysis statistics not inflated.

In our analysis, we have used publicly available migraine GWAS from FinnGen R8, which is done according to the best practices of the FinnGen project. Replacing that GWAS with a new one would require redoing all our analyses starting from the meta-analysis, the locus definitions, the LD matrix generations, fine-mapping and the results reported in Figures and Tables. We note that while the intercept value seems somewhat elevated from the null value of 1.0, the LDSC plots (Supplementary Figure 1) very clearly show that the association signal is overwhelmingly due to polygenicity.

To conservatively account for the observed elevation in the intercept, we have now followed the LDSC paper (Bulik-Sullivan 2015 Nat Genet)⁶, where they suggested that the LDSC intercept can be used as a genomic control parameter that divides the chi-square association statistics before the P-values are computed. We now report for all new loci in Table 2 whether the lead SNP remains genome-wide significant ($P < 5 \times 10^{-8}$) in our analysis after the LDSC-intercept correction (division of chi-square statistic by 1.09). We explain this approach in Methods p.32

The intercept of the LD score regression analysis can be used as a genomic control parameter to provide a conservative P-value by dividing the observed chi-square statistic of the GWAS association by the intercept before computing the P-value from the chi-square distribution⁶. We used this approach to indicate which variants in Table 2 had $P > 5 \times 10^{-8}$ after the genomic control.

and report in the main text (p.7):

After genomic control by the LDSC intercept, 22 novel loci remained GWS (Table 2).

5) For the validation of new loci from meta-analysis, the significance level should be 0.05/35, or FDR with threshold 0.05 can be used. Either way, it is important to adjust for number of tests.

One locus, *IPO8* ($P = 0.0003$), is validated by the replication data at the significance level 0.05/35. We have added this information in the main text, p.8.

*Of the 35 lead variants of our new loci, 32 were consistent in direction ($P = 2.1 \times 10^{-7}$, one-sided binomial test), 17 replicated with $P < 0.05$ (one-sided test; Supplementary Table 2) and the *IPO8* locus was validated at $P < 0.05/35$ (one-sided test) in the replication data.*

Please note that given the small effect sizes of the migraine risk variants, the replication data we have been able to evaluate here is not able to provide strong statistical evidence for any one locus on its own, but the combined signal in the sign test is clear (32 out of 35 being consistent in direction between the discovery and replication).

6) Table 2. It is better to report odds ratio instead of log-odds ratio because odds ratio is easier to understand its effect size.

We have changed the column “log-odds ratio” to “odds ratio” and column “S.e.” to “95%CI” with definition in the legend as “95%CI = 95% confidence interval”. We have done corresponding changes also in Table 3.

1. Isgut, M., Song, K., Ehm, M.G., Wang, M.D. & Davitte, J. Effect of case and control definitions on genome-wide association study (GWAS) findings. *Genetic Epidemiology* **47**, 394-406 (2023).
2. Uffelmann, E. *et al.* Genome-wide association studies. *Nature Reviews Methods Primers* **1**, 59 (2021).
3. Hautakangas, H. *et al.* Genome-wide analysis of 102,084 migraine cases identifies 123 risk loci and subtype-specific risk alleles. *Nature Genetics* **54**, 152-160 (2022).
4. Pickrell, J.K. *et al.* Detection and interpretation of shared genetic influences on 42 human traits. *Nature Genetics* **48**, 709-717 (2016).
5. Kurki, M.I. *et al.* FinnGen provides genetic insights from a well-phenotyped isolated population. *Nature* **613**, 508-518 (2023).
6. Bulik-Sullivan, B. *et al.* LD Score regression distinguishes confounding from polygenicity in genome-wide association studies. *Nature genetics* **47**, 291-295 (2015).

7. Trubetsky, V. *et al.* Mapping genomic loci implicates genes and synaptic biology in schizophrenia. *Nature* **604**, 502-508 (2022).
8. Meyers, T.J. *et al.* Transcriptome-wide association study identifies novel candidate susceptibility genes for migraine. *Human Genetics and Genomics Advances* **4**(2023).
9. Bahrami, S. *et al.* Dissecting the shared genetic basis of migraine and mental disorders using novel statistical tools. *Brain* **145**, 142-153 (2021).
10. Guo, D.-c. *et al.* Genetic Variants in *LRP1* and *ULK4* Are Associated with Acute Aortic Dissections. *The American Journal of Human Genetics* **99**, 762-769 (2016).
11. Georges, A. *et al.* Genetic investigation of fibromuscular dysplasia identifies risk loci and shared genetics with common cardiovascular diseases. *Nature Communications* **12**, 6031 (2021).
12. Turley, T.N. *et al.* Identification of Susceptibility Loci for Spontaneous Coronary Artery Dissection. *JAMA Cardiology* **5**, 929-938 (2020).
13. Bjornsdottir, G. *et al.* Rare variants with large effects provide functional insights into the pathology of migraine subtypes, with and without aura. *Nature Genetics* **55**, 1843-1853 (2023).
14. Korsching, S., Auburger, G., Heumann, R., Scott, J. & Thoenen, H. Levels of nerve growth factor and its mRNA in the central nervous system of the rat correlate with cholinergic innervation. *The EMBO Journal* **4**, 1389-1393 (1985).
15. Connor, B. & Dragunow, M. The role of neuronal growth factors in neurodegenerative disorders of the human brain. *Brain Research Reviews* **27**, 1-39 (1998).
16. van Dongen, R.M. *et al.* Migraine biomarkers in cerebrospinal fluid: A systematic review and meta-analysis. *Cephalalgia* **37**, 49-63 (2017).
17. Mozafarihashjin, M. *et al.* Assessment of peripheral biomarkers potentially involved in episodic and chronic migraine: a case-control study with a focus on NGF, BDNF, VEGF, and PGE2. *The Journal of Headache and Pain* **23**, 3 (2022).
18. Aloe, L., Tuveri, M.A., Carcassi, U. & Levi-Montalcini, R. Nerve growth factor in the synovial fluid of patients with chronic arthritis. *Arthritis & Rheumatism* **35**, 351-355 (1992).
19. Freemont, A.J. *et al.* Nerve growth factor expression and innervation of the painful intervertebral disc. *The Journal of Pathology* **197**, 286-292 (2002).
20. Walsh, D.A. *et al.* Angiogenesis and nerve growth factor at the osteochondral junction in rheumatoid arthritis and osteoarthritis. *Rheumatology (Oxford)* **49**, 1852-61 (2010).
21. Sanga, P. *et al.* Efficacy, safety, and tolerability of fulranumab, an anti-nerve growth factor antibody, in the treatment of patients with moderate to severe osteoarthritis pain. *Pain* **154**, 1910-1919 (2013).
22. Tiseo, P.J., Ren, H. & Mellis, S. Fasinumab (REGN475), an antinerve growth factor monoclonal antibody, for the treatment of acute sciatic pain: results of a proof-of-concept study. *Journal of Pain Research* **7**, 523-30 (2014).
23. Watt, F.E. *et al.* Tropomyosin-related kinase A (TrkA) inhibition for the treatment of painful knee osteoarthritis: results from a randomized controlled phase 2a trial. *Osteoarthritis Cartilage* **27**, 1590-1598 (2019).

24. Berenbaum, F. *et al.* Subcutaneous tanezumab for osteoarthritis of the hip or knee: efficacy and safety results from a 24-week randomised phase III study with a 24-week follow-up period. *Annals of the Rheumatic Diseases* **79**, 800-810 (2020).
25. Wise, B.L., Seidel, M.F. & Lane, N.E. The evolution of nerve growth factor inhibition in clinical medicine. *Nature Reviews Rheumatology* **17**, 34-46 (2021).
26. Koya, Y. *et al.* A novel anti-NGF PEGylated Fab' provides analgesia with lower risk of adverse effects. *mAbs* **15**, 2149055 (2023).
27. Detke, H.C. *et al.* Galcanezumab in chronic migraine: The randomized, double-blind, placebo-controlled REGAIN study. *Neurology* **91**, e2211-e2221 (2018).
28. Dodick, D.W. *et al.* ARISE: A Phase 3 randomized trial of erenumab for episodic migraine. *Cephalalgia* **38**, 1026-1037 (2018).
29. Ferrari, M.D. *et al.* Fremanezumab versus placebo for migraine prevention in patients with documented failure to up to four migraine preventive medication classes (FOCUS): a randomised, double-blind, placebo-controlled, phase 3b trial. *The Lancet* **394**, 1030-1040 (2019).
30. De Smedt, F., Verjans, B., Mailleux, P. & Erneux, C. Cloning and expression of human brain type I inositol 1,4,5-trisphosphate 5-phosphatase High levels of mRNA in cerebellar Purkinje cells. *FEBS Letters* **347**, 69-72 (1994).
31. Zorzetto, M. *et al.* SERPINA1 Gene Variants in Individuals from the General Population with Reduced α 1-Antitrypsin Concentrations. *Clinical Chemistry* **54**, 1331-1338 (2008).
32. Storck, S.E., Kurtyka, M. & Pietrzik, C.U. Brain endothelial LRP1 maintains blood-brain barrier integrity. *Fluids and Barriers of the CNS* **18**, 27 (2021).
33. Liu, Z., Andraska, E., Akinbode, D., Mars, W. & Alvidrez, R.I.M. LRP1 in the Vascular Wall. *Current Pathobiology Reports* **10**, 23-34 (2022).
34. Lee, J. *et al.* ANKS1A regulates LDL receptor-related protein 1 (LRP1)-mediated cerebrovascular clearance in brain endothelial cells. *Nature Communications* **14**, 8463 (2023).
35. Liu, L. *et al.* LRP1 Repression by SNAIL Results in ECM Remodeling in Genetic Risk for Vascular Diseases. *Circulation Research* **135**, 1084-1097 (2024).
36. Choquet, H. *et al.* New and sex-specific migraine susceptibility loci identified from a multiethnic genome-wide meta-analysis. *Communications Biology* **4**, 864 (2021).
37. Gupta, R.M. *et al.* A Genetic Variant Associated with Five Vascular Diseases Is a Distal Regulator of Endothelin-1 Gene Expression. *Cell* **170**, 522-533.e15 (2017).
38. Wang, X. & Musunuru, K. Confirmation of Causal rs9349379-PHACTR1 Expression Quantitative Trait Locus in Human-Induced Pluripotent Stem Cell Endothelial Cells. *Circulation: Genomic and Precision Medicine* **11**, e002327 (2018).
39. Rubin, S. *et al.* PHACTR-1 (Phosphatase and Actin Regulator 1) Deficiency in Either Endothelial or Smooth Muscle Cells Does Not Predispose Mice to Nonatherosclerotic Arteriopathies in 3 Transgenic Mice. *Arteriosclerosis, Thrombosis, and Vascular Biology* **42**, 597-609 (2022).
40. Anttila, V. *et al.* Genome-wide association study of migraine implicates a common susceptibility variant on 8q22.1. *Nature Genetics* **42**, 869-873 (2010).

41. Chasman, D.I. *et al.* Genome-wide association study reveals three susceptibility loci for common migraine in the general population. *Nature Genetics* **43**, 695-U116 (2011).
42. Freilinger, T. *et al.* Genome-wide association analysis identifies susceptibility loci for migraine without aura. *Nature Genetics* **44**, 777-782 (2012).
43. Anttila, V. *et al.* Genome-wide meta-analysis identifies new susceptibility loci for migraine. *Nature Genetics* **45**, 912-U255 (2013).
44. Gormley, P. *et al.* Meta-analysis of 375,000 individuals identifies 38 susceptibility loci for migraine. *Nature Genetics* **48**, 856-866 (2016).
45. Chen, S.-P. *et al.* Genome-wide association study identifies novel susceptibility loci for migraine in Han Chinese resided in Taiwan. *Cephalalgia* **38**, 466-475 (2018).
46. Chang, X. *et al.* Common variants at 5q33.1 predispose to migraine in African-American children. *Journal of Medical Genetics* **55**, 831 (2018).
47. Pasaniuc, B. *et al.* Fast and accurate imputation of summary statistics enhances evidence of functional enrichment. *Bioinformatics* **30**, 2906-2914 (2014).
48. Wu, Y., Eskin, E. & Sankararaman, S. A Unifying Framework for Imputing Summary Statistics in Genome-Wide Association Studies. *Journal of Computational Biology* **27**, 418-428 (2020).
49. Cai, M. *et al.* XMAP: Cross-population fine-mapping by leveraging genetic diversity and accounting for confounding bias. *Nature Communications* **14**, 6870 (2023).
50. Zou, Y., Carbonetto, P., Wang, G. & Stephens, M. Fine-mapping from summary data with the “Sum of Single Effects” model. *PLOS Genetics* **18**, e1010299 (2022).

REVIEWERS' COMMENTS

Reviewer #1 (Remarks to the Author):

The authors have provided a comprehensive and well-structured response, and the additional analyses substantially strengthen the manuscript. Most original concerns are addressed, and the data appear robust. A few points remain:

Major

1. Details on Bonferroni correction and significance thresholds improve clarity. However, the PheWAS section uses lenient thresholds, with results proposed as “for further screening.” Please indicate in the main text or supplement how many associations remain significant under stricter global corrections (e.g., Bonferroni or FDR). Has the Bonferroni threshold accounted for correlations among analyses, and could a more flexible FDR approach improve power?

We thank the reviewer for this helpful suggestion. As the reviewer mentioned, Bonferroni correction would be unsuitable due to complex and unknown correlations among the phenotypes. With an FDR approach (Benjamini–Hochberg method), all reported associations remained significant at 0.05 level across the three scans. We have now added this information to the main text on pages 14,15.

... we identified 404 variant-disease associations with $P < 1 \times 10^{-5}$ (Supplementary Data 6, https://hhautakangas.github.io/phewas_migraine_tables.html). All associations remained significant at a false discovery rate (FDR) of 0.05.

... and identified 122 variant-disease associations with $P < 1 \times 10^{-4}$ (Supplementary Data 8, https://hhautakangas.github.io/phewas_migraine_tables.html), including traits such as sleep apnea and stroke. All associations remained significant at an FDR of 0.05.

...we identified 330 variant-disease associations with $P < 1 \times 10^{-4}$ (Supplementary Data 9, https://hhautakangas.github.io/phewas_migraine_tables.html), including, e.g., focal epilepsy and hydrocephalus. All associations remained significant at an FDR of 0.05.

We also added FDR adjusted *P*-value columns to the PheWAS results in Supplementary Data 6, 8, and 9.

2. QC was conducted separately in the three cohorts, then harmonized using unified filters (e.g., MAF, INFO, HWE). While reasonable, could excluding multi-allelic variants remove functionally important rare variants? Please report the proportion excluded and discuss implications for fine-mapping reliability.

We thank the reviewer for this important point. Multi-allelic sites were excluded during harmonization, which could in principle remove functionally relevant rare variants. We have discussed potential problems related to excluding rare variants and multi-allelic variants on page 19:

Additionally, since very rare variants or multi-allelic variants were not included in our analysis, we miss the true causal variants that are rare or multi-allelic. Since our variant set is not comprehensive, we must keep in mind that also variants that have a very high probability of being causal in our analysis may still have such variants in high LD that were not included in our analysis. A valid calibration of the PIPs would require that all potential causal variants were included in the analysis. In practice, for common variants, this would require comprehensively imputed data sets with no missing variants in any of the meta-analyzed studies, and, for rare variants, availability of high coverage sequencing data. Currently, we do not yet have such resources available in typical GWAS meta-analyses of common diseases such as migraine.

The proportion of multi-allelic sites was very small in all three cohorts: 0.2% in 23andMe, 0.5% in UK Biobank, and 0.8% in FinnGen. Given these low proportions, the impact on downstream analyses, including fine-mapping, is expected to be small.

We have added the details of the proportions excluded to the revised manuscript on page 26:

We excluded multi-allelic variants (0.2% in 23andMe, 0.5% in UK Biobank, and 0.8% in FinnGen), and variants with minor allele frequency (MAF) < 0.01, IMPUTE2 info or MACH r^2 < 0.6, and when available, missingness > 0.05 and Hardy-Weinberg equilibrium (HWE) $P < 1 \times 10^{-6}$ from each study.

Minor

1. LDSC intercepts, LD scores, and PENC thresholds were added to control confounding, and the absence of rare variants was acknowledged. After LDSC adjustment, which novel loci showed the greatest loss of significance? Could some be driven by residual confounding?

After applying the LDSC intercept adjustment to control for potential confounding, 13 loci no longer exceeded the genome-wide significance threshold. The original P -values for these loci ranged from $4.56e-08$ to $1.41e-08$, and the LDSC-adjusted P -values ranged from $1.60e-07$ to $5.41e-08$.

The three loci showing the greatest loss of significance were:

- near *FTHL17* (P -value shifted from $4.56e-08$ to $1.60e-07$),
- near *COX19* (from $4.40e-08$ to $1.54e-07$), and
- near *DTL* (from $3.71e-08$ to $1.32e-07$).

While some loss of significance after the LDSC adjustment may indicate that residual confounding contributed to the original signals, most genome-wide significant loci remained robust to adjustment. We therefore conclude that confounding is unlikely to be a major driver of the novel findings.

2. The explanation for lower genetic correlation in FinnGen and its limited effect on effect size estimates is convincing. Still, a sensitivity analysis excluding FinnGen (UKB+23andMe only) for key loci would further test robustness.

To address this, we have provided forest plots for all loci in Supplementary Figure 3. As shown, FinnGen does not exhibit systematic bias relative to 23andMe and UK Biobank.

We also report heterogeneity values in Supplementary Data 2. Only 2 loci showed significant heterogeneity (reported on page 8), indicating that the lower genetic correlation observed in FinnGen is unlikely to have affected the overall findings.

p.8:

*We observed statistically significant heterogeneity ($P < 0.05/122$) in effect sizes between the study collections only for two lead variants, both of which resided in the previously known migraine loci (*PRDM16* and near *ZCCHC14*) (Supplementary Data 1, Supplementary Fig 3).*

Reviewer #2 (Remarks to the Author):

The authors have done an excellent job addressing my comments.

We sincerely thank you for your kind words and for recognizing our efforts in addressing your comments.

Reviewer #3 (Remarks to the Author):

We appreciate the time you dedicated to co-reviewing our manuscript. Thank you for your contribution to improving our work.

Reviewer #4 (Remarks to the Author):

The authors have addressed my questions adequately. I have no other questions.

We are grateful for your feedback and are pleased that our revisions have adequately addressed your questions.